# On the Spectral Bias of Convolutional Neural Tangent and Gaussian Process Kernels

Amnon Geifman[1]         Meirav Galun[1]         David Jacobs[2]         Ronen Basri[1]

[1]Department of Computer Science, Weizmann Institute of Science, Rehovot, Israel
`{amnon.geifman, meirav.galun, ronen.basri}@weizmann.ac.il`

[2]Department of Computer Science, University of Maryland, College Park, MD
`djacobs@cs.umd.edu`

## Abstract

We study the properties of various over-parameterized convolutional neural architectures through their respective Gaussian Process and Neural Tangent kernels. We prove that, with normalized multi-channel input and ReLU activation, the eigenfunctions of these kernels with the uniform measure are formed by products of spherical harmonics, defined over the channels of the different pixels. We next use hierarchical factorizable kernels to bound their respective eigenvalues. We show that the eigenvalues decay polynomially, quantify the rate of decay, and derive measures that reflect the composition of hierarchical features in these networks. Our theory provides a concrete quantitative characterization of the role of locality and hierarchy in the inductive bias of over-parameterized convolutional network architectures.

## 1   Introduction

Convolutional Neural Networks (CNNs) [23] have produced dramatic improvements over past machine learning approaches [22, 35, 18]. Two key properties that distinguish CNNs are their ability to encode geometric properties of the data, by incorporating multiscale analysis and invariance or equivariance. Shift invariant networks produce the same output when the input is shifted, which can be valuable, for example, in classification tasks in which objects are not well aligned. Shift equivariant networks produce shifted output when the input is shifted, and are important in image-to-image networks that, for example, denoise or segment the input [21, 24, 37]. Multiscale feature representations naturally arise in these networks as the receptive field size varies with depth.

However, we still lack a theoretical analysis of CNNs that can quantitatively predict their behavior. Our analysis builds on the Gaussian Process and Neural Tangent kernels (resp. GPK and NTK). It has been shown theoretically that massively overparameterized networks can be well approximated by a linearization about their initialization [1, 2, 15, 20]. With this linearization, neural networks become kernel regressors, with training dynamics and smoothness properties determined by the eigenfunctions and eigenvalues of their kernel, which determine their Reproducing Kernel Hilbert Space (RKHS).

A series of interesting works has derived the spectrum of NTK for fully connected networks (denoted FC-NTK). This eigen-analysis tells us which functions a network learns most rapidly, since the speed of learning an eigenfunction with gradient descent (GD) is inversely proportional to the corresponding eigenvalue [6]. For example, it allows us to determine that FC-NTK learns low frequency components of a function faster than high frequency components, and characterize the rates at which this happens [4, 5, 6, 8, 9]. So this eigen-analysis allows us to characterize the inductive bias of over-parametrized neural networks. Convolutional GPKs and NTKs (resp. CGPKs and CNTKs) have been derived for

36th Conference on Neural Information Processing Systems (NeurIPS 2022).

convolutional networks [2, 30], but a characterization of their inductive bias is still missing, despite several recent works that provide only limited characterizations (see review in Sec. 4).

In this paper we investigate the Gaussian Process and Neural Tangent kernels associated with three deep convolutional architectures. In particular, we consider kernels associated with a shift equivariant architecture, a convnet in which the last layer is fully connected, and a convnet with a final global average pooling step. The former network is applicable to various image processing tasks. The second network is similar in architecture to AlexNet and VGG [22, 35]. The latter network resembles a residual network [18], without skip connections. All models we consider use ReLU activation.

We assume our networks receive multi-channel input signals, with the channels for each pixel normalized to unit norm. Our results establish that the eigenfunctions of the kernels include either products of spherical harmonics (SH-products) or sums of SH-products that are invariant to shifting the input. The eigenvalues of these kernels decay polynomially with the frequency of the eigenfunctions with a leading coefficient that is determined by the hierarchical architecture of the corresponding CNN. Consequently, for the equivariant network the eigenvalues are larger for functions that are localized in the center of the receptive field. For the other two networks the eigenvalues are larger for functions that are spatially localized (in any position in the input image). This spectral analysis implies that like FC-networks, CNNs are biased to learn low-frequency target functions more quickly. Unlike FC-networks, they can more quickly learn high frequency functions when these are variable in a small neighborhood of pixels. (Note that high frequency reflects high variability of the target function for similar input images and does not refer to the power spectrum of individual images.)

## 2 Preliminaries and notations

We consider a multi-channel 1-D input signal $\mathbf{x}$ of length $d$ (referred to as *pixels*), each with $\zeta$ channels. We represent $\mathbf{x}$ by a $\zeta \times d$ matrix and denote the $i$'th pixel by $\mathbf{x}^{(i)} \in \mathbb{R}^\zeta$, $i \in [d]$. We further set $D = \zeta d$. We note that our results can readily be applied also to 2-D, multi-channel signals. We assume further that the entries of each pixel are normalized to unit norm, i.e., $\left\| \mathbf{x}^{(i)} \right\| = \left\| \left( x_1^{(i)}, \ldots, x_\zeta^{(i)} \right) \right\| = 1$. The input space, therefore, is a Cartesian product of spheres, which we call *multisphere*, i.e., $\mathbf{x} \in \mathbb{MS}(\zeta, d) = \underbrace{\mathbb{S}^{\zeta-1} \times \ldots \times \mathbb{S}^{\zeta-1}}_{d} \subset \sqrt{d}\mathbb{S}^{D-1}$ (i.e., a sphere with radius $\sqrt{d}$).

We use multi-index notations, i.e., $\mathbf{n} = (n_1, \ldots, n_d), \mathbf{k} = (k_1, \ldots, k_d) \in \mathbb{N}^d$, to denote vectors of polynomial orders or frequencies. $\mathbb{N}$ denotes the natural numbers including zero, and $b_\mathbf{n}, \lambda_\mathbf{k} \in \mathbb{R}$ are scalars that depend on vectors of indices $\mathbf{n}$ or $\mathbf{k}$. We denote monomials by $\mathbf{t}^\mathbf{n} = t_1^{n_1} t_2^{n_2} \cdot \ldots \cdot t_d^{n_d}$ with $\mathbf{t} \in \mathbb{R}^d$, and allow also for a scalar exponent, i.e., $\mathbf{t}^n = (t_1 \cdot \ldots \cdot t_d)^n$. For $\mathbf{u}, \mathbf{v} \in \mathbb{R}^d$ we say that $\mathbf{u} \geq \mathbf{v}$ if $u_i \geq v_i$ for all $i \in [d]$. Therefore, the power series $\sum_{\mathbf{n} \geq \mathbf{0}} b_\mathbf{n} \mathbf{t}^\mathbf{n}$ should read $\sum_{n_1 \geq 0, n_2 \geq 0, \ldots} b_{n_1, n_2, \ldots} t_1^{n_1} t_2^{n_2} \ldots$

We denote by $s_i \mathbf{x} = (\mathbf{x}^{(i+1)}, \ldots \mathbf{x}^{(d)}, \mathbf{x}^{(1)}, \ldots \mathbf{x}^{(i)})$ the cyclic shift of $\mathbf{x}$ to the left by $i$ pixels. We denote the uniform distribution in a domain $\Omega$ by $\text{Unif}(\Omega)$. We write $f(x) \sim g(x)$ when $\lim_{x \to \infty} f(x)/g(x) = 1$. Throughout the paper we assume all kernels are differentiable at zero infinitely many times and their power series converge in the hypercube $[-1, 1]^d$. Our theorems and lemmas are proved in the supplementary material.

### 2.1 The network model

We consider convolutional neural network architectures (Figure 1) of the following form. Given a multi-channel 1-D input signal $\mathbf{x} \in \mathbb{MS}(\zeta, d)$ arranged in a $\zeta \times d$ matrix, we use a shift equivariant backbone and three heads to produce scalar features. The network includes $L \geq 2$ layers. The first layer implements a $1 \times 1$ convolution layer. This is followed by $L - 1$ stride-1 convolutional layers with filters of size $q$, producing at each layer the same number of feature channels $m$ in each of the $d$ locations.

The $f^{\text{Eq}}$ head implements a fully convolutional network; it produces one scalar entry for the shift equivariant network (i.e., the tuple $(f^{\text{Eq}}(\mathbf{x}, \cdot), \ldots, f^{\text{Eq}}(s_{d-1}\mathbf{x}, \cdot))$ produces the shift-equivariant output). $f^{\text{Tr}}$ corresponds to applying a fully connected layer at the top layer. Finally, $f^{\text{GAP}}$ corresponds

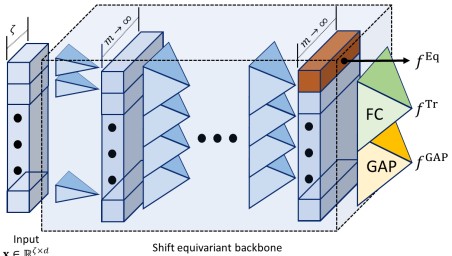

$$\begin{aligned}
&1.\ \mathbf{f}^{(1)}(\mathbf{x},\theta)=\sigma\left(W^{(1)}\mathbf{x}\right).\\[4pt]
&2.\ \mathbf{f}_i^{(l)}(\mathbf{x},\theta)=\\
&\quad\sigma\left(\sqrt{\tfrac{c_\sigma}{mq}}\left(\sum_{j=1}^m W^{(l)}_{:,i,j}*\mathbf{f}_j^{(l-1)}(\mathbf{x},\theta)\right)\right),\\[4pt]
&3.\ \text{with } l\in\{2,\dots,L\},\, i\in[m],\text{ and three heads:}\\[4pt]
&\quad(a)\ f^{\mathrm{Eq}}(\mathbf{x},\theta)=\langle \mathbf{w}^{\mathrm{Eq}},\mathbf{f}^{(L)}_{:,1}(\mathbf{x},\theta)\rangle.\\[4pt]
&\quad(b)\ f^{\mathrm{Tr}}(\mathbf{x},\theta)=\tfrac{1}{\sqrt{d}}\langle W^{\mathrm{Tr}},\mathbf{f}^{(L)}(\mathbf{x},\theta)\rangle.\\[4pt]
&\quad(c)\ f^{\mathrm{GAP}}(\mathbf{x},\theta)=\tfrac{1}{d}\mathbf{w}^{\mathrm{GAP}}\mathbf{f}^{(L)}(\mathbf{x},\theta)\mathbf{1}.
\end{aligned}$$

Figure 1: Left: Network architecture. An input signal $\mathbf{x}\in\mathbb{R}^{\zeta\times d}$ (left column) is fed into an equivariant backbone (dashed box), producing $m$ feature channels for each pixel, first using $1\times1$ convolution, followed by $L-1$ convolution layers with filters of size $q$ (marked by blueish pyramids), interleaved with ReLU activation. The backbone is followed by one of three heads, $f^{\mathrm{Eq}}$, $f^{\mathrm{Tr}}$ and $f^{\mathrm{GAP}}$. Right: The corresponding formulas. Here $\mathbf{f}^{(l)}\in\mathbb{R}^{m\times d}$ $(l\in[L])$; $\theta=\left(W^{(1)},...,W^{(L)},W^{\mathrm{Tr}},\mathbf{w}^{\mathrm{GAP}},\mathbf{w}^{\mathrm{Eq}}\right)$ are learnable parameters initialized with $\mathcal{N}(0,I)$. $W^{(1)}\in\mathbb{R}^{m\times\zeta}$, $W^{(l)}\in\mathbb{R}^{q\times m\times m}$ (i.e., $W^{(l)}_{:,i,j}$ is a filter of size $q$), $W^{\mathrm{Tr}}\in\mathbb{R}^{m\times d}$ ($\langle\cdot,\cdot\rangle$ denotes the standard inner product between matrices), $\mathbf{w}^{\mathrm{Eq}},\mathbf{w}^{\mathrm{GAP}}\in\mathbb{R}^{1\times m}$ and $\mathbf{1}=(1,...,1)^T\in\mathbb{R}^d$. '$*$' denotes cyclic convolution; $\sigma$ is the ReLU function, and for ReLU, $c_\sigma=1/\left(\mathbb{E}_{z\sim\mathcal{N}(0,1)}[\sigma(z)^2]\right)=2$.

Table 1: CGPK and CNTK formulas. Given $\mathbf{x},\mathbf{z}\in\mathbb{MS}(\zeta,d)$, we denote respectively by $X$ and $Z$ their $\zeta\times d$ matrix representations. Here we denote by $\tilde{\Sigma}$, $\tilde{\dot{\Sigma}}$, and $\tilde{\Theta}$ respectively $\Sigma$, $\dot{\Sigma}$ and $\Theta$ whose rows and columns are extended with circular padding. Additionally, with ReLU activation $\kappa_0(u)=1-\arccos(u)/\pi$, and $\kappa_1(u)=u\kappa_0(u)+\sqrt{1-u^2}/\pi,\quad u\in[-1,1].$

Let $\Sigma^{(0)}(\mathbf{x},\mathbf{z})=\Theta^{(0)}(\mathbf{x},\mathbf{z})=X^T Z$. For $l\in[L]$ and $i,j\in[d]$,

1. $\Sigma^{(1)}_{i,j}(\mathbf{x},\mathbf{z})=\kappa_1\left(\Sigma^{(0)}_{i,j}(\mathbf{x},\mathbf{z})\right).$

2. $\dot{\Sigma}^{(1)}_{i,j}(\mathbf{x},\mathbf{z})=\kappa_0\left(\Sigma^{(0)}_{i,j}(\mathbf{x},\mathbf{z})\right).$

3. $\Theta^{(l)}_{i,j}(\mathbf{x},\mathbf{z})=\frac{1}{2q}\sum_{r=0}^{q-1}\left[\tilde{\dot{\Sigma}}^{(l)}_{i+r,j+r}(\mathbf{x},\mathbf{z})\tilde{\Theta}^{(l-1)}_{i+r,j+r}(\mathbf{x},\mathbf{z})+\tilde{\Sigma}^{(l)}_{i+r,j+r}(\mathbf{x},\mathbf{z})\right].$

4. $\Sigma^{(l+1)}_{i,j}(\mathbf{x},\mathbf{z})=\kappa_1\left(\frac{1}{q}\sum_{r=0}^{q-1}\tilde{\Sigma}^{(l)}_{i+r,j+r}(\mathbf{x},\mathbf{z})\right).$

5. $\dot{\Sigma}^{(l+1)}_{i,j}(\mathbf{x},\mathbf{z})=\kappa_0\left(\frac{1}{q}\sum_{r=0}^{q-1}\tilde{\Sigma}^{(l)}_{i+r,j+r}(\mathbf{x},\mathbf{z})\right),$

1. CGPK-EqNet (corresponds to $f^{\mathrm{Eq}}$): $\Sigma^{(L)}_{1,1}(\mathbf{x},\mathbf{z}).$

2. CGPK (corr. to $f^{\mathrm{Tr}}$): $\frac{1}{d}\sum_{i=1}^d\Sigma^{(L)}_{i,i}(\mathbf{x},\mathbf{z}).$

3. CGPK-GAP (corr. to $f^{\mathrm{GAP}}$): $\frac{1}{d^2}\sum_{i=1}^d\sum_{j=1}^d\Sigma^{(L)}_{i,j}(\mathbf{x},\mathbf{z}).$

4. CNTK-EqNet (corr. to $f^{\mathrm{Eq}}$): $\Theta^{(L)}_{1,1}(\mathbf{x},\mathbf{z}).$

5. CNTK (corr. to $f^{\mathrm{Tr}}$): $\frac{1}{d}\sum_{i=1}^d\Theta^{(L)}_{i,i}(\mathbf{x},\mathbf{z}).$

6. CNTK-GAP (corr. to $f^{\mathrm{GAP}}$): $\frac{1}{d^2}\sum_{i=1}^d\sum_{j=1}^d\Theta^{(L)}_{i,j}(\mathbf{x},\mathbf{z}).$

to applying global average pooling, resulting in a shift invariant network. With each of the heads, the network parameters are trained for regression with the mean square error (MSE) loss.

## 2.2 Derivation of CGPK and CNTK

Previous work showed that in the limit of infinite width and with small initialization, massively over-parameterized neural networks become kernel regressors with a family of kernels called Neural Tangent Kernels [20]. Let $f(\mathbf{x},\theta)$ denote a network, then for a pair of inputs $\mathbf{x}_i,\mathbf{x}_j$, the corresponding NTK is defined as $\mathbf{k}(\mathbf{x}_i,\mathbf{x}_j)=\mathbb{E}_{\theta\sim\mathcal{P}}\left\langle\frac{\partial f(\mathbf{x}_i,\theta)}{\partial\theta},\frac{\partial f(\mathbf{x}_j,\theta)}{\partial\theta}\right\rangle$. A related kernel, called the Gaussian Process (or random feature) kernel, is obtained if the weights are kept in their initial random values, and only the last layer of the network is optimized in training [13].

[2] derived expressions for CNTK and CGPK for convolutional networks and noted that they can be computed for a pair of inputs in $O(d^2L)$. The formulas in Table 1 adapt these expressions to our convolutional architectures and to multisphere inputs. We note that with general input in $\mathbb{R}^D$ additional normalization steps are needed. We refer the reader to [2] for the full derivation.

Note that for a pair of inputs $\mathbf{x},\mathbf{z}$, this definition produces two matrices of kernels, $\Sigma^{(L)}_{i,j}(\mathbf{x},\mathbf{z})$ and $\Theta^{(L)}_{i,j}(\mathbf{x},\mathbf{z})$, $i,j\in[d]$, and that $\Sigma^{(L)}_{i,j}(\mathbf{x},\mathbf{z})=\Sigma^{(L)}_{1,1}(s_{i-1}\mathbf{x},s_{j-1}\mathbf{z})$ (and similarly for $\Theta$). With these definitions, we produce six different kernels in Table 1(right)–these describe three different architectures for each of the Gaussian Process and the Neural Tangent kernels.

Below we refer to these six kernels as CGPKs and CNTKs. Note that CGPK-EqNet and CNTK-EqNet each produce a single output, corresponding to the first output of the equivariant network. The tuple $(\boldsymbol{k}(s_0\mathbf{x}, s_0\mathbf{z}), ..., \boldsymbol{k}(s_{d-1}\mathbf{x}, s_{d-1}\mathbf{z})) \in \mathbb{R}^d$ (with $\boldsymbol{k}$ either CGPK-EqNet or CNTK-EqNet) produces the full response of the equivariant network.

# 3 The RKHS of CGPKs and CNTKs

Our objective is to derive the spectrum of Gaussian Process and Neural Tangent kernels associated with convolutional networks. We do so by forming bounds using products of kernels that apply to individual pixels and are composed hierarchically. Prior work on FC-NTK has benefited from the fact that they are dot product kernels, allowing the analysis to draw on a rich body of results. Our task is challenging since the kernels we study are not dot product kernels; tools for spectral decomposition of these kernels on the sphere are scarcely available. We approach this problem by first proving general results for kernels that are functions of inner products between pixels (Sec. 3.1). We next consider factorizable kernels and derive their spectrum (Sec. 3.2). Then, in Sec. 3.3, we refine our expressions to account for hierarchical kernels. We finally use these derivations in Sec. 3.4 and 3.5 to prove bounds for all six kernels.

## 3.1 Multi-dot product kernels

It can be readily shown that CNTK and CGPK associated with the shift equivariant network are functions of dot products of corresponding pixels. We refer to such kernels as *multi-dot product* and prove in the supplementary material several new results regarding their spectral properties, briefly summarized here.

We call a kernel $\boldsymbol{k}(\mathbf{x}, \mathbf{z}) : \mathbb{MS}(\zeta, d) \times \mathbb{MS}(\zeta, d) \to \mathbb{R}$ *multi-dot product* if $\boldsymbol{k}(\mathbf{x}, \mathbf{z}) = \boldsymbol{k}(\mathbf{t})$, where $\mathbf{t} = (\langle \mathbf{x}^{(1)}, \mathbf{z}^{(1)} \rangle, .., \langle \mathbf{x}^{(d)}, \mathbf{z}^{(d)} \rangle) \in [-1, 1]^d$. (Note the overload of notation, which should be clear by context.) Multi-dot product kernels can be written via Mercer's decomposition as

$$\boldsymbol{k}(\mathbf{x}, \mathbf{z}) = \sum_{\mathbf{k}, \mathbf{j}} \lambda_{\mathbf{k}} Y_{\mathbf{k}, \mathbf{j}}(\mathbf{x}) Y_{\mathbf{k}, \mathbf{j}}(\mathbf{z}), \tag{1}$$

where $Y_{\mathbf{k}, \mathbf{j}}(\mathbf{x})$ $(\mathbf{k}, \mathbf{j} \in \mathbb{N}^d)$, the eigenfunctions of $\boldsymbol{k}$, are products of spherical harmonics in $\mathbb{S}^{\zeta-1}$, $Y_{\mathbf{k}, \mathbf{j}}(\mathbf{x}) = \prod_{i=1}^{d} Y_{k_i j_i}(\mathbf{x}^{(i)})$, with $k_i \geq 0$, $j_i \in [N(\zeta, k_i)]$, and $N(\zeta, k_i)$ denotes the number of harmonics of frequency $k_i$ in $\mathbb{S}^{\zeta-1}$. Such products are harmonic polynomials in $\mathbb{S}^{D-1}$. With $\zeta = 2$, these are products of Fourier series in a multi-dimensional torus. The eigenvalues $\lambda_{\mathbf{k}}$ depend on the vector of frequencies, $\mathbf{k}$, and are independent of the phases $\mathbf{j}$. We note that a multi-dot product kernel is universal for $\mathbb{MS}(\zeta, d)$ if all its eigenvalues are strictly positive. Non-universal kernels are obtained, e.g., when the eigenfunctions involve pixels outside of the receptive field of the respective network, in which case these eigenfunctions lie in the null space of the kernel.

Below we consider kernels that can be expressed using a multivariate power series of the form

$$\boldsymbol{k}(\mathbf{t}) = \sum_{\mathbf{n} \geq \mathbf{0}} b_{\mathbf{n}} \mathbf{t}^{\mathbf{n}} = \sum_{\mathbf{n} \geq \mathbf{0}} b_{\mathbf{n}} \prod_{i=1}^{d} \langle \mathbf{x}^{(i)}, \mathbf{z}^{(i)} \rangle^{n_i}. \tag{2}$$

with $b_{\mathbf{n}} \geq 0$ for all $\mathbf{n} \geq \mathbf{0}$. It can be readily shown that our CGPKs and CNTKs are positive semidefinite (PSD) and their power series coefficients are non-negative; the kernels are obtained from the univariate PSD $\kappa_0$ and $\kappa_1$ (defined in Table 1), whose coefficients are non-negative, by sequences of multiplication, addition and composition, resulting in PSD kernels with non-negative coefficients.

The eigenvalues of multi-dot product kernels can be deduced from their power series coefficients. This is established in the following lemma, which extends a result by [3, 32] to multi-dot product kernels.

**Lemma 3.1.** *Let $\boldsymbol{k}$ be a multi-dot product kernel with the power series given in (2), where $\mathbf{x}^{(i)}, \mathbf{z}^{(i)} \in \mathbb{S}^{\zeta-1}$ respectively are pixels in $\mathbf{x}, \mathbf{z}$. Then, the eigenvalues $\lambda_{\mathbf{k}}(\boldsymbol{k})$ of $\boldsymbol{k}$ are given by*

$$\lambda_{\mathbf{k}}(\boldsymbol{k}) = \left| \mathbb{S}^{\zeta-2} \right|^d \sum_{\mathbf{s} \geq 0} b_{\mathbf{k}+2\mathbf{s}} \prod_{i=1}^{d} \lambda_{k_i}(t^{k_i + 2s_i}),$$

*where $|\mathbb{S}^{\zeta-2}|$ is the surface area of $\mathbb{S}^{\zeta-2}$, and $\lambda_k(t^n)$ is the k'th eigenvalue of $t^n$, given by*

$$\lambda_k(t^n) = \frac{n!}{(n-k)!2^{k+1}} \frac{\Gamma\left(\frac{\zeta-1}{2}\right)\Gamma\left(\frac{n-k+1}{2}\right)}{\Gamma\left(\frac{n-k+\zeta}{2}\right)}$$

*if $n - k$ is even and non-negative, while $\lambda_k(t^n) = 0$ otherwise, and $\Gamma$ is the Gamma function.*

Lemma 3.1 generalizes a result by [32] (Thm. 4.1), which deals with the special case of a kernel made of a single convolutional layer followed by $L$ FC layers. Our result applies to any multi-dot product kernel, including convolutional kernels of arbitrary depths such as CNTK and CGPK.

## 3.2 Factorizable Kernels

Next, we consider multivariate kernels that factor into products of dot product kernels, $\bar{k}(\mathbf{x}, \mathbf{z}) = \prod_{i=1}^{d} \bar{k}_i(\langle \mathbf{x}^{(i)}, \mathbf{z}^{(i)} \rangle)$. The power series of $\bar{k}$ can be written as $\bar{k}(\mathbf{t}) = \prod_{i=1}^{d} \left(\sum_{n=0}^{\infty} b_n^{(i)} t_i^n\right)$. Their eigenvalues satisfy $\lambda_{\mathbf{k}}(\bar{k}) = \prod_{i=1}^{d} \lambda_{k_i}(\bar{k}_i)$ and can be calculated using Lemma 3.1. Below we are interested specifically in kernels whose power series decay polynomially with frequency. The next theorem shows that for such kernels, the eigenvalues too decay polynomially and derives their exact decay rate. For the theorem we further use the concept of a receptive field, which captures for a multivariate kernel the subset of variables it depends on. That is, the receptive field $\mathcal{R} \subseteq [d]$ is the set of indices $i$ for which there exists $\mathbf{n} = (..., n_i, ...)$ with $n_i \geq 1$ and $b_{\mathbf{n}} \neq 0$. This condition ensures that there exists a term in the power series expansion that depends on pixel $i$.

**Theorem 3.2.** *Let $\bar{k}$ be a factorizable multi-dot product kernel with inputs $\mathbf{x}, \mathbf{z} \in \mathbb{MS}(\zeta, d)$, and let $\mathcal{R} \subseteq [d]$ denote its receptive field. Suppose that $\bar{k}$ can be written as a multivariate power series, $\bar{k}(\mathbf{t}) = \sum_{\mathbf{n} \geq 0} b_{\mathbf{n}} \mathbf{t}^{\mathbf{n}}$ with $b_{\mathbf{n}} \sim c \prod_{i \in \mathcal{R}, n_i > 0} n_i^{-\nu}$. with constants $c > 0$, non-integer $\nu > 1$, and $b_{\mathbf{n}} = 0$ if $n_i > 0$ for any $i \notin \mathcal{R}$. Then the eigenfunctions of $\bar{k}$ w.r.t the uniform measure are the SH-products. Moreover, let $\mathbf{k} \in \mathbb{N}^d$ be a vector of frequencies. Then, the eigenvalues $\lambda_{\mathbf{k}}(\bar{k})$ satisfy*

$$\lambda_{\mathbf{k}} \sim \tilde{c} \prod_{i \in \mathcal{R}, k_i > 0} k_i^{-(\zeta+2\nu-3)}.$$

*Finally, $\lambda_{\mathbf{k}} = 0$ if $k_i > 0$ for any $i \notin \mathcal{R}$.*

The theorem above significantly improves over previous results by [3, 33] in various ways. Specifically, these authors considered only dot product kernels (i.e., $d = 1$) and bounded the decay of their eigenvalues by a non-tight upper bound. Here we provide both tight upper and lower bounds, which are identical up to a constant factor, for any $d \geq 1$.

## 3.3 Positional bias of eigenvalues

CNNs process signals by producing a hierarchy of learned features that emerge via repeated convolutions, interleaved with non-linear activations. It is natural to ask, therefore, to what extent this hierarchy is reflected in the RKHS of the CGPKs and CNTKs. To answer this question we consider kernels formed by hierarchical composition of kernels. We are further interested in such kernels that are both factorizable and whose power series decay polynomially. For such kernels we prove that their eigenvalues are larger for eigenfunctions that depend on pixels near the center of their receptive field and smaller for eigenfunctions that depend only on peripheral pixels. This in turn will allow us to derive similar bounds for the kernels associated with the equivariant network. For the trace and GAP kernels this will reveal bias to eigenfunctions that depend on nearby pixels, compared to those that depend only on more distant pixels.

**Definition 3.3.** A kernel $\bar{k}^{(L)} : [-1, 1]^d \to \mathbb{R}$ is called (stride-1) hierarchical of depth $L > 1$ and filter size $q$ if there exists a sequence of kernels $\bar{k}^{(1)}, ..., \bar{k}^{(L)}$ such that $\bar{k}^{(l)}(\mathbf{t}) = f^{(l)}\left(\bar{k}^{(l-1)}(s_0\mathbf{t}), ..., \bar{k}^{(l-1)}(s_{q-1}\mathbf{t})\right)$ with $f^{(l)} : \mathbb{R}^q \to \mathbb{R}$ and $\bar{k}^{(1)}(\mathbf{t}) = f^{(1)}(t_1), t_1 \in [-1, 1]$.

Similar to feed-forward networks, hierarchical kernels induce a tree structure in which the leaf nodes represent the variables $t_1, ..., t_d$ and parent nodes represent function applications $f^{(l)}$. Each variable

may be connected by multiple paths to the root node, which corresponds to the final kernel $\boldsymbol{k}^{(L)}$. The number of paths from a node $t_i$ to the root, denoted $p_i^{(L)}$, plays an important role in the magnitude of the eigenvalues. In the next theorem we bound both the power series coefficients and the eigenvalues of hierarchical kernels, showing that the bounds depend on the number of paths in the hierarchical tree, which in turn is determined by the position of the pixel within the receptive field.

**Theorem 3.4.** *Let $\boldsymbol{k}^{(L)} = \sum_{\mathbf{n} \geq 0} b_{\mathbf{n}} \mathbf{t}^{\mathbf{n}}$ be hierarchical and factorizable of depth $L > 1$ with filter size $q$ so that $b_{\mathbf{n}} = c \prod_{i=1, n_i > 0}^{d} n_i^{-\nu}$ for non-integer $\nu > 1$. Then there exists a scalar $A > 1$ s.t.:*

1. *The power series coefficients of $\boldsymbol{k}^{(L)}$ satisfy $b_{\mathbf{n}} \geq c_L \prod_{\substack{i=1 \\ n_i > 0}}^{d} A^{\min(p_i^{(L)}, n_i)} n_i^{-\nu}$.*

2. *The eigenvalues $\lambda_{\mathbf{k}}(\boldsymbol{k}^{(L)})$ satisfy $\lambda_{\mathbf{k}} \geq \tilde{c}_L \prod_{\substack{i=1 \\ k_i > 0}}^{d} A^{\min(p_i^{(L)}, k_i)} k_i^{-(\zeta + 2\nu - 3)}$,*

*where $c_L, \tilde{c}_L$ are constants that depend on $L$, and $p_i^{(L)}$ denotes the number of paths from pixel $i$ to the output of $\boldsymbol{k}^{(L)}$.*

The proof exploits relations between hierarchical stride-1 and stride-$q$ kernels and relies on recursively combining power series, which are shown to maintain their polynomial decay. We note that although the number of paths can grow rapidly with depth, the kernels we consider are normalized so that they always produce values in $[-1, 1]$. As a result, both the power series coefficients and the eigenvalues are bounded in $[0, 1]$. This normalization is reflected in the magnitudes of $c_L$ and $\tilde{c}_L$.

In fact, for a hierarchical, stride-1 kernel of depth $L$, the (normalized) number of paths $p_i^{(L)}$ of a pixel, $\mathbf{x}^{(i)}$, decays exponentially with its distance from the center of the receptive field, $|i - i_c|$. The number of paths is determined by convolving a rectangular function of width $q$, the size of the convolution filter, (i.e., $r(x) = 1/q$ if $0 \leq x \leq q$ and 0 otherwise) with itself $L$ times. Note that such a repeated convolution is equal to the density function obtained by summing $L$ random variables distributed uniformly, resulting in the Irwin–Hall distribution, whose support is stretched by the size of the convolution filter $q$. With sufficiently large $L$, using the central limit theorem and assuming the number of pixels $d$ is greater than the receptive field size, the number of paths $p_i^{(L)}$ approaches a Gaussian centered at the central pixel with variance $V_{q,L} \approx Lq^2/12$, i.e., $p_i^{(L)} \propto \exp(-(i - i_c)^2/(2V_{q,L}))$ [25]. Roughly speaking, we conclude that the position of a pixel within the receptive field has an exponential effect on both the power series coefficients and the eigenvalues of the kernel.

Note that in the analysis above we assumed that the number of pixels $d$ is larger than the receptive field size. With a receptive field of size $d$ and cyclic convolutions, the number of paths from all pixels approaches a constant as $L$ grows. In practice, real CNNs use zero padding, which effectively ensures that $d$ is always larger than the receptive field size.

## 3.4 Kernels associated with the equivariant network

The following theorem characterizes the spectrums of CGPK-EqNet and CNTK-EqNet, by proving lower and upper bounds on both their power series coefficients as well as their eigenvalues.

**Theorem 3.5.** *Let $\boldsymbol{k}^{(L)}$ denote either CGPK-EqNet or CNTK-EqNet of depth $L$ whose input includes $\zeta$ channels, with receptive field $\mathcal{R}$ and with ReLU activation. Then,*

1. *$\boldsymbol{k}^{(L)}$ can be written as a power series, $\boldsymbol{k}^{(L)}(\mathbf{t}) = \sum_{\mathbf{n} \geq 0} b_{\mathbf{n}} \mathbf{t}^{\mathbf{n}}$ with*

$$c_1 \prod_{i \in \mathcal{R}, n_i > 0} \tilde{A}^{\min(p_i^{(L)}, n_i)} n_i^{-2.5} \leq b_{\mathbf{n}} \leq c_2 \prod_{i \in \mathcal{R}, n_i > 0} n_i^{-\nu},$$

2. *The eigenvalues of $\boldsymbol{k}^{(L)}$ are bounded by*

$$c_3 \prod_{\substack{i \in \mathcal{R} \\ k_i > 0}} \tilde{A}^{\min(p_i^{(L)}, k_i)} k_i^{-(\zeta + 2)} \leq \lambda_{\mathbf{k}} \leq c_4 \prod_{\substack{i \in \mathcal{R} \\ k_i > 0}} k_i^{-(\zeta + 2\nu - 3)},$$

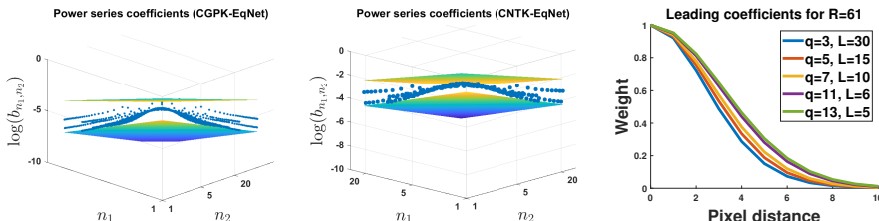

Figure 2: The power series coefficients of CGPK-EqNet (left, $n \leq 50$) and CNTK-EqNet (middle, $n \leq 20$). Here $d = 2$, $q = 2$, $L = 2$ and the receptive field size is 2. The coefficients, $b_{\mathbf{n}}$, marked with blue points, are shown as functions of $\mathbf{n} = (n_1, n_2)$. Our bounds are depicted in each graph by two planes whose slopes are determined by the exponents in Theorem 3.5. Right: The figure shows the relative magnitude of the eigenvalues of a hierarchical trace kernel for SH-products involving two pixels as a function of the distance between the pixels. This is shown for various architectures with different depths $L$ and convolution filter sizes $q$ that results in the same receptive field size of 61.

*where for CGPK-EqNet $\nu = 1 + 3/(2d)$ and for CNTK-EqNet $\nu = 1 + 1/(2d)$. Also, $p_i^{(L)}$ denotes the number of paths from pixel $i$ to the output of $\bar{\mathbf{k}}^{(L)}$, $\tilde{A} > 1$ and $c_1, c_2, c_3$ and $c_4$ depend on $L$.*

The proof relies on the recursive formulation of the kernels, which gives rise to a multivariate version of the Faá di Bruno formula [34], involving Bell polynomials; those are exploited to bound the coefficients of the power series expansion of the kernels. The implication of Theorem 3.5 is that CGPK-EqNet and CNTK-EqNet are bounded from above and below by factorizable kernels. The eigenvalues of these bounding kernels decay polynomially with the product of the frequency in each pixel, $k_i$, with an exponent that depends on $\zeta$, the number of input channels. While this implies in general that with GD, low frequencies are learned faster than high frequencies, the theorem also states that it should be faster to learn target functions that vary (i.e., have high frequencies) in a small set of pixels compared to ones that vary in many pixels.

Recall that learning an eigenfunction with eigenvalue $\lambda$ requires $O(1/\lambda)$ GD iterations (e.g., [6]). Therefore, for example, according to the lower bound in Thm. 3.5, learning an SH-product of frequency $k$ in one pixel and constant frequencies in the other pixels should require $O(k^{\zeta+2})$ GD iterations. Learning an SH-product of frequency $k/\bar{m}$ in each of $\bar{m}$ pixels should require $O((k/\bar{m})^{\bar{m}(\zeta+2)})$ iterations. With $k \gg \bar{m}$, this is exponentially slower by a factor of $\bar{m}$. With $\bar{m} = d$, the speed of learning decays with an exponent that depends on the size of the entire signal. For such functions, over-parameterized CNNs behave similarly to fully connected networks, for which the eigenvalues of their respective kernels, FC-GPK and FC-NTK, decay as resp. $k^{-(D+2)}$ and $k^{-D}$ for input in $\mathbb{S}^{D-1}$ [8, 17, 12]. Note that the exponent in the polynomial decay does not depend on $q$, the size of the convolution filter. We conclude that over-parameterized CNNs can more efficiently learn target functions whose variation is restricted to subsets of the pixels. This is not true for fully connected networks. We further illustrate this difference between convolutional and FC kernels in Sec. 3.5, where we numerically compare the eigenvalues for the trace kernels.

While the form of the polynomial decay induces bias toward learning functions that depend on fewer pixels, according to the lower bound the multiplicative factors of these polynomials exhibit bias toward learning functions in which the variation is localized near the center of the receptive field. This bias is affected by the number of paths from a pixel node to the output and depends on the depth and the size of the convolution filter $q$.

Figure 2 provides a visualization of the power series coefficients of CGPK-EqNet and CNTK-EqNet for $d = 2$ and $q = 2$. The upper and lower bounds, plotted as planes in these log-log plots, indicate the maximal and minimal asymptotic directions determined by the exponents in Theorem 3.5. The coefficients indeed appear to lie between the bounds, with coefficients of orders that vary along a single axis (i.e., $(n, 0)$) lying close to the lower bound, while those of orders that vary simultaneously along both axes (i.e., $(n, n)$) lie close to the upper bound. In this example the number of paths are equal for both pixels, so position does not affect the coefficients.

## 3.5 Trace and GAP kernels

Our next objective is to characterize the RKHS of CGPK and CNTK and their GAP versions.

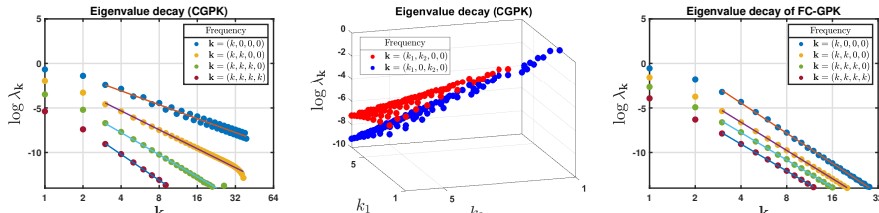

Figure 3: Left: The eigenvalues of CGPK for frequency patterns that include either one non zero frequency (blue dots), two (orange), three (green) or four (maroon) identical frequencies. The slopes (respectively, $-5.18$, $-7.06$, $-8.24$ and $-9.64$) indicate the exponent for each pattern. Middle: The eigenvalues of CGPK for two frequency patterns that include exactly two non zero frequency, either next to each other (red dots) or separated by one zero frequency (blue). Consistent with our theory, the eigenvalues obtained with non-zero frequencies next to each other are larger than those obtained with the same frequencies but separated. Right: The eigenvalues of FC-GPK on $\mathbb{MS}(\zeta, d)$ for frequency patterns that include either one non zero frequency (blue dots), two (orange), three (green) or four (maroon) identical frequencies. The slopes (respectively, $-11.26$, $-10.52$, $-10.20$ and $-10.04$) indicate the exponent for each pattern. We used $d = 4$, $\zeta = 3$, $q = 2$, $L = 3$ in these plots.

**Definition 3.6.** Let $\bar{k}(\mathbf{x}, \mathbf{z})$ be a multi-dot product kernel. We define the respective trace kernel by $\boldsymbol{k}^{\mathrm{Tr}}(\mathbf{x}, \mathbf{z}) = \frac{1}{d} \sum_{i=0}^{d-1} \bar{k}(s_i \mathbf{x}, s_i \mathbf{z})$ and GAP kernel by $\boldsymbol{k}^{\mathrm{GAP}}(\mathbf{x}, \mathbf{z}) = \frac{1}{d^2} \sum_{i=0}^{d-1} \sum_{j=0}^{d-1} \bar{k}(s_i \mathbf{x}, s_j \mathbf{z})$.

Clearly, by their definition (Section 2.2), CGPK and CNTK respectively are the trace kernels of CGPK-EqNet and CNTK-EqNet, while CGPK-GAP and CNTK-GAP are their GAP versions.

The eigenvalues and eigenfunctions of trace and GAP kernels can be derived from their generating kernel, as is established in the following lemma.

**Theorem 3.7.** *Let $\bar{k}$ be a multi-dot-product kernel with Mercer's decomposition as in* (1)*, and let $\boldsymbol{k}^{\mathrm{Tr}}$ and $\boldsymbol{k}^{\mathrm{GAP}}$ respectively be its trace and GAP versions. Then,*

1. *$\boldsymbol{k}^{\mathrm{Tr}}(\mathbf{x}, \mathbf{z}) = \sum_{\mathbf{k}, \mathbf{j}} \lambda_{\mathbf{k}}^{\mathrm{Tr}} Y_{\mathbf{k}, \mathbf{j}}(\mathbf{x}) Y_{\mathbf{k}, \mathbf{j}}(\mathbf{z})$ with $\lambda_{\mathbf{k}}^{\mathrm{Tr}} = \frac{1}{d} \sum_{i=0}^{d-1} \lambda_{s_i \mathbf{k}}$, where $\lambda_{\mathbf{k}}$ denotes an eigenvalue of $\bar{k}$.*

2. *$\boldsymbol{k}^{\mathrm{GAP}}(\mathbf{x}, \mathbf{z}) = \sum_{\mathbf{k}, \mathbf{j}} \lambda_{\mathbf{k}}^{\mathrm{Tr}} \tilde{Y}_{\mathbf{k}, \mathbf{j}}(\mathbf{x}) \tilde{Y}_{\mathbf{k}, \mathbf{j}}(\mathbf{z})$ with $\tilde{Y}_{\mathbf{k}, \mathbf{j}}(\mathbf{x}) = \frac{1}{\sqrt{d}} \sum_{i=0}^{d-1} Y_{s_i \mathbf{k}, s_i \mathbf{j}}(\mathbf{x})$.*

According to this theorem, the eigenfunctions of a trace kernel are the SH-products. This is simply because the trace kernel itself is multi-dot product. Its eigenvalues are obtained by averaging the eigenvalues of $\bar{k}$ for shifted frequencies $s_i \mathbf{k}$, making the eigenvalues $\lambda_{\mathbf{k}}$ invariant to shifts of the index vector $\mathbf{k}$. Note however that for a trace kernel, the respective eigenfunctions are not shift invariant. The second part of the lemma establishes that GAP kernels share the same eigenvalues as their respective trace kernels. For these kernels, however, the eigenfunctions consist of sums of SH-products that are shift invariant.

By combining Thms 3.7 and 3.5 we obtain the following bounds on the eigenvalues of these kernels.

**Corollary 3.8.** *Let $\bar{k}^{(L)}$ denote either CGPK, CGPK-GAP, CNTK or CNTK-GAP of depth $L$ whose input includes $\zeta$ channels, with receptive field $\mathcal{R}$ and with ReLU activation. Then, the eigenvalues of $\bar{k}^{(L)}$ are bounded by*

$$\sum_{j=0}^{d-1} c_3 \prod_{i \in \mathcal{R}, k_{i+j} > 0} \tilde{A}^{\min(p_{i+j}^{(L)}, k_{i+j})} k_{i+j}^{-(\zeta+2)} \leq \lambda_{\mathbf{k}} \leq \sum_{j=0}^{d-1} c_4 \prod_{i \in \mathcal{R}, k_{i+j} > 0} k_{i+j}^{-(\zeta+2\nu-3)},$$

*where for CGPK, CGPK-GAP $\nu = 1 + 3/(2d)$, while for CNTK, CNTK-GAP $\nu = 1 + 1/(2d)$. Also, $p_i^{(L)}$ is defined in Theorem 3.5, i.e., the number of paths in the **equivariant kernel**, $\tilde{A} > 1$ and $c_3$ and $c_4$ depend on $L$. Here $k_{i+j}$ is identified with $i + j - d$ if $i + j > d$.*

Overall, Corollary 3.8 indicates that the eigenvalues of the trace and GAP CGPK and CNTK decay polynomially with bounds that are similar to those of the equivariant kernel. Here too, the exponent depends on the number of channels and forms a bias toward learning target functions that depend

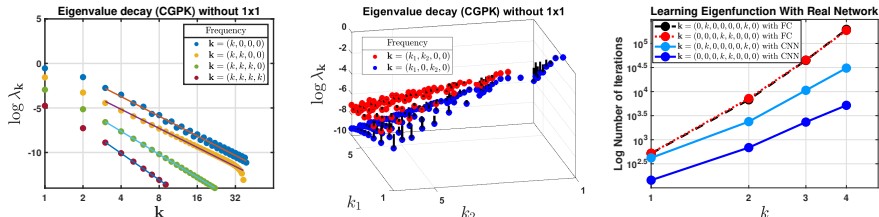

Figure 4: Left: The eigenvalues of CGPK without the $1 \times 1$ convolution. Eigenvalues for different frequency patterns that include either one nonzero frequency (blue dots), two (orange), three (green) or four (maroon) identical frequencies. The slopes (respectively, $-7.32, -7.11, -8.58, -9.89$) indicate the exponent for each pattern. Middle: Eigenvalues for two frequency patterns that include exactly two nonzero frequencies, either next to each other (red dots) or separated by one zero frequency (blue). Black lines indicate differences from the values shown in Figure 3. ($d = 4, \zeta = 2, q = 2, L = 3$.) Right: Convergence times of real networks, a CNN (solid dark and light blue curves) and a fully connected network (dashed black and dotted red), trained to fit various SH-products. ($d = 8, \zeta = 2, q = 2, L = 4$.)

on few pixels. The multiplicative coefficients depend on the number of paths from each pixel to the output in the corresponding equivariant kernel. In contrast to the equivariant kernels, the eigenvalues of the trace and GAP kernels are averages over shifted indices, and therefore they depend on the *relative* positions of pixels. For example, the eigenvalues of SH-products involving only two pixels with large frequencies, $k_i$ and $k_{i+\delta}$, is large when $\delta$ is small, and is smaller when $\delta$ is large. This is illustrated in the graph in Figure 2(right), which shows the relative magnitude of the eigenvalues, according to the lower bound, involving exactly two pixels as a function of their distance. It can be seen that the coefficients decay exponentially with the distance between the two pixels. Moreover, for a fixed receptive field size, a more rapid decay is obtained with a smaller filter and deeper architecture, compared to a larger filter and shallower network.

The graphs in Figure 3(left and middle) show the eigenvalues of the CGPK kernel evaluated numerically for a three-layer network. The figure shows the eigenvalues for frequencies $k$ in 1–4 pixels. The slope of these lines, which depicts the exponent of the decay of the corresponding eigenvalues, is in good correspondence with our theory. Moreover, consistent with our discussion in Sec. 3.4, the decay rate is higher for frequencies spread over more pixels. This can be contrasted with the eigenvalues for the Gaussian Process Kernel for a fully connected network (FC-GPK) shown in Figure 3(right). For that kernel, the decay rate is the same regardless of the pixel spread of frequencies, as is indicated by the parallel lines in the graph, and is similar to the maximal decay rate obtained for CGPK with $d = 4$ pixels. Figure 3(middle) further shows the eigenvalues of CGPK obtained with two frequency patterns, $(k_1, k_2, 0, 0, ...)$ and $(k_1, 0, k_2, 0, ...)$. As is predicted by our theory, while the decay rate for both patterns is similar, the eigenvalues for high frequencies in two adjacent pixels are larger (by a multiplicative constant) than for the same frequencies in two non-adjacent ones. This is of course not the case with FC-NTK, in which the spatial location of pixels makes no difference.

**Removing the $1 \times 1$ convolution**: Our analysis uses network models which include a $1 \times 1$ convolution in the first layer. This simplifies our derivations and appears to have only limited effect on the analysis. In particular, it can be readily shown that for data distributed uniformly on the multisphere, the eigenfunctions of CGPK and CNTK without the $1 \times 1$ convolution are the SH-products (or their shift invariant sums in the case of GAP kernels). Figure 4(left and middle) shows the eigenvalues of CGPK under the same conditions as in Figure 3 with the $1 \times 1$ convolution removed. Overall, similar decay patterns are observed, except that with $q = 2$ in the case of high frequency in one pixel (denoted '$(k, 0, 0, 0)$') the eigenvalues decay faster than with the $1 \times 1$ convolution.

**Real networks**: We further tested our predictions on two real networks, a CNN and a fully connected architecture. We trained the networks to fit various SH-products in which the higher frequencies ($k = 1, ..., 4$) are positioned at either two neighboring or two distant pixels and measured the number of iterations to convergence. Figure 4(right) shows that, consistent with our predictions, the CNN learned each function faster than the FC network. Moreover, the CNN learned the localized functions faster than those in which the high frequencies are spread in distant pixels. The FC network, in contrast, required the same number of iterations to converge for both types of functions.

# 4 Related Work

A number of papers analyze the spectral properties and generalization bounds for convolutional kernels, including CNTK. [9] derived explicit feature maps for CNTK. [29] developed spectral decomposition and generalization bounds for CNTK with only one convolution layer and pooling with input in the nodes of the hyper-cube $\{-1, 1\}^d$. [16] considered CNTK with just one convolutional layer that is applied to an input comprising of non-overlapping patches. Such input is equivalent to applying only a $1 \times 1$ convolution layer in our setup. They showed that the eigenvalues decay for this kernel is dictated by the patch size (corresponding to the number of channels in our setup). Their limited setup yields Mercer's decomposition similar to that obtained with NTK for a two-layer FC network.

[28] considered a network with one convolution layer with patches of full image size and inputs drawn from the uniform distribution on either the sphere or the hypercube. Their work focuses on the benefits of group invariance in reducing the sample size. [10] considered general kernels that incorporate group invariance and derived a generalization bound based on counting the number of eigenfunctions of the kernel. [38] studied the spectral properties of CNTK in a setting in which the frequency is held constant, while the input dimension tends to infinity. Subsequent to our work, [11] derived asymptotic decay rates of the eigenvalues of CNTK and further used these to derive a new generalization bound. Both of these recent works, however, are restricted to networks that involve convolutions with *non-overlapping* patches.

[27, 26] proposed a convolutional kernel network (CKN) that involves layers of patch extraction, convolution, and pooling. Pooling in this model is implemented by a convolution with a fixed filter such as a Gaussian, possibly followed by subsampling. We note that without subsampling, such pooling operation can effectively result in a fully connected layer. [32] proved that the set of functions produced by the CKN network that applies convolution with non-overlapping patches is contained in the RKHS of a kernel that includes just one convolution layer. Similar to our work, their work used inputs drawn from a multisphere, yielding eigenfunctions that are SH-products. Their spectral analysis however was applied to CKNs with only one convolution layer followed by a sequence of polynomial activations. [7] extended this setup to enable convolution with overlapping patches and further derive generalization bounds. In contrast to these works, our work shows that the SH-products are the eigenfunctions of all multi-dot product kernels and further provides spectral bounds for CGPK and CNTK for multilayer CNNs.

[19] investigated the speed of convergence of GD for equivariant networks with a fully connected layer followed by a fixed convolution layer in the context of image denoising. In a related work, [36] used NTK for a two-layer network to relate denoising with CNNs to denoising with non-local filters. Existing work also investigates CNNs from a non-kernel perspective, for example, by showing that multi-layer CNNs (and related networks) can efficiently learn *compositional* functions [31, 14].

# 5 Conclusion

Our paper has derived the eigenfunctions and corresponding eigenvalues of Neural Tangent Kernels that describe overparameterized CNNs. This provides us with a clear and specific understanding of these networks, just as much prior work has done for fully connected networks. Of particular interest, our work provides a concrete understanding of the inductive bias produced by the hierarchical structure of deep CNNs. We see that CNNs can efficiently learn higher frequency functions than FC networks when these functions are spatially localized. We provide formulas that can show the trade-off points between higher frequency functions and spatial localization, and also show how architectural variations can affect these biases. We believe that this is a significant step towards understanding which features CNNs will learn, and how this depend on network architecture. In future work we hope to provide tighter bounds for the eigenvalues, extend our analysis to residual networks, and further test our predictions on real convolutional networks.

**Acknowledgement**. This research was partially supported by the Israeli Council for Higher Education (CHE) via the Weizmann Data Science Research Center, by research grants from the Estate of Tully and Michele Plesser and the Anita James Rosen Foundation, and by the U.S.-Israel Binational Science Foundation, grant no. 2018680. DJ is further supported by the National Science Foundation, grant no. IIS-1910132 and DARPA's Guaranteeing AI Robustness Against Deception (GARD).

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
