# – Supplementary Material –
# On the Spectral Bias of Convolutional Neural Tangent and Gaussian Process Kernels

Amnon Geifman[1]        Meirav Galun[1]        David Jacobs[2]        Ronen Basri[1]

[1]Department of Computer Science, Weizmann Institute of Science, Rehovot, Israel

`{amnon.geifman, meirav.galun, ronen.basri}@weizmann.ac.il`

[2]Department of Computer Science, University of Maryland, College Park, MD

`djacobs@cs.umd.edu`

## A    Multi-dot product kernels

In this section we prove results presented in Section 3.1.

**Lemma A.1.** *CGPK-EqNet and CNTK-EqNet are multi-dot product kernels.*

*Proof.* The proof follows directly from the derivation of the kernels in Section 2.2. Note that CGPK-EqNet is given by $\tilde{\Sigma}_{11}^{(L)}$ and CNTK-EqNet by $\tilde{\Theta}_{11}^{(L)}$, and their recursive definition only involve elements of $\tilde{\Sigma}^{(l)}$ and $\tilde{\Theta}^{(l)}$ in which $i = j$. Moreover, by definition the diagonal elements of $\Sigma^{(0)}$ and $\Theta^{(0)}$ are $\langle \mathbf{x}^{(i)}, \mathbf{z}^{(i)} \rangle$ for $i \in [d]$, implying the lemma. $\qquad\square$

### A.1    Multivariate Gegenbauer Polynomials

We next extend basic results derived for functions on the sphere to the multisphere $\mathbb{MS}(\zeta, d)$. These results will assist us later to prove Mercer's decomposition for multi-dot product kernels in the subsequent section.

We consider the set of Gegenbauer polynomials $\{Q_k^{(\zeta)}(t)\}_{k\geq 0}$ that are orthogonal in $L_2[-1, 1]$ w.r.t. the weight function $(1 - t^2)^{(\zeta-3)/2}$ and omit the superscript. Inspired by [4], we define multivariate Gegenbauer polynomials, using facts from harmonic analysis on the sphere. (See references [6, 7] for background on spherical harmonics and Gegenbauer polynomials). We denote by $|\mathbb{S}^{\zeta-1}|$ the area of the sphere $\mathbb{S}^{\zeta-1}$.

**Definition A.2.** For $k \geq 0$, let $Q_k(t) : [-1, 1] \to \mathbb{R}$ be the (univariate) Gegenbauer polynomial of degree $k$. Then, the multivariate Gegenbauer polynomial of order $\mathbf{k}$ is $Q_{\mathbf{k}}(\mathbf{t}) : [-1, 1]^d \to \mathbb{R}$, defined by

$$Q_{\mathbf{k}}(\mathbf{t}) = Q_{k_1}(t_1) \cdot Q_{k_2}(t_2) \cdot \ldots \cdot Q_{k_d}(t_d).$$

These multivariate Gegenbauer polynomials enjoy several properties that they inherit from their univariate counterpart.

**Lemma A.3.** *Let $P_k(\mathbf{t})$ denote the space of polynomials of degree $\leq k$ with variables $\mathbf{t} \in [-1, 1]^d$. Then, the set $\{Q_{\mathbf{i}}(\mathbf{t})\}_{\mathbf{i}=0}^{|\mathbf{i}|=k}$ is an orthogonal basis of $P_k(\mathbf{t})$ w.r.t. the weight function $(1 - t^2)^{(\zeta-3)/2}$ (with $\mathbf{i} = (i_1, \ldots, i_d)$ and $|\mathbf{i}| = i_1 + \ldots + i_d$).*

*Proof.* Let $p(\mathbf{t}) = \sum_{\mathbf{i}=0}^{|\mathbf{i}|=k} a_{\mathbf{i}} \mathbf{t}^{\mathbf{i}} \in P_k(\mathbf{t})$. Since the univariate Gegenbauer polynomials form an orthogonal basis, for every $0 \leq n_i \leq k$ and $i \in [d]$ we can write $t_i^{n_i} = \sum_{j=0}^{n_i} a_j^{(n_i)} Q_j(t_i)$, where

$a_j^{(n_i)} \in \mathbb{R}$, and the superscript is used to emphasize that the expansion depends on $n_i$. Therefore, $p(\mathbf{t})$ can be written as

$$p(\mathbf{t}) = \sum_{\mathbf{n}=0}^{|\mathbf{n}|=k} a_{\mathbf{n}} \mathbf{t^n} = \sum_{\mathbf{n}=0}^{|\mathbf{n}|=k} a_{\mathbf{n}} \left( \sum_{j=0}^{n_1} a_j^{(n_1)} Q_j(t_1) \right) \cdot \ldots \cdot \left( \sum_{j=0}^{n_d} a_j^{(n_d)} Q_j(t_d) \right)$$

$$=^{(1)} \sum_{\mathbf{n}=0}^{|\mathbf{n}|=k} \tilde{a}_{\mathbf{n}} Q_{n_1}(t_1) Q_{i_2}(t_2) \cdot \ldots \cdot Q_{n_d}(t_d) = \sum_{\mathbf{n}=0}^{|\mathbf{n}|=k} \tilde{a}_{\mathbf{n}} Q_{\mathbf{n}}(\mathbf{t}),$$

where $^{(1)}$ is obtained by applying the distributive law with the fact that $n_1, .., n_d \leq k$. finally, $\tilde{a}_{\mathbf{i}}$ can be computed explicitly from $a_{\mathbf{i}}$ and $\{a_j^{(n_i)}\}$.

We have shown that $P_k(\mathbf{t})$ is spanned by the set $\{Q_{\mathbf{i}}(\mathbf{t})\}_{\mathbf{i}=0}^{|\mathbf{i}|=k}$. Next, we show that this set is orthogonal with respect to the measure $\prod_{r=1}^{d} \left( (1-t_r^2)^{\frac{\zeta-3}{2}} \right)$. Let $\mathbf{i}$ and $\mathbf{j}$ be two vectors of indices. Then, we have that

$$\int_{[-1,1]^d} Q_{\mathbf{i}}(\mathbf{t}) Q_{\mathbf{j}}(\mathbf{t}) \prod_{r=1}^{d} (1-t_r^2)^{\frac{\zeta-3}{2}} dt_1 \cdot \ldots \cdot dt_d = \prod_{r=1}^{d} \left( \int_{[-1,1]} Q_{i_r}(t_r) Q_{j_r}(t_r) (1-t_r^2)^{\frac{\zeta-3}{2}} dt_r \right)$$

$$= \left( \frac{|\mathbb{S}^{\zeta-1}|}{|\mathbb{S}^{\zeta-2}|} \right)^d \left( \prod_{r=1}^{d} N(\zeta, i_r) \right)^{-1} \delta_{i_1,j_1} \cdot \delta_{i_2,j_2} \cdot \ldots \cdot \delta_{i_d,j_d},$$

where the last equality is due to the orthogonality property of the univariate Gegenbauer polynomials. This concludes the proof. $\square$

The relation of the multivariate Gegenbauer polynomials to the SH-products is formulated in the following lemma.

**Lemma A.4.** *Let* $\mathbf{x}, \mathbf{z} \in \mathbb{MS}(\zeta, d)$. *It holds that*

$$Q_{\mathbf{k}}(\langle \mathbf{x}^{(1)}, \mathbf{z}^{(1)} \rangle, .., \langle \mathbf{x}^{(i)}, \mathbf{z}^{(j)} \rangle, .., \langle \mathbf{x}^{(d)}, \mathbf{z}^{(d)} \rangle) = |\mathbb{S}^{\zeta-1}|^d \left( \prod_{r=1}^{d} N(q, k_r) \right)^{-1} \sum_{\mathbf{j}: j_r \in [N(\zeta, k_r)]} Y_{\mathbf{k},\mathbf{j}}(\mathbf{x}) Y_{\mathbf{k},\mathbf{j}}(\mathbf{z}),$$

*where* $Y_{\mathbf{k},\mathbf{j}}(\mathbf{x})$ *is homogeneous polynomial of degree* $k_1 + .. + k_d$. $Y_{\mathbf{k},\mathbf{j}}(\mathbf{x})$ *is further given by SH-products, i.e.,* $Y_{\mathbf{k},\mathbf{j}}(\mathbf{x}) = \prod_{i=1}^{d} Y_{k_i,j_i}(\mathbf{x}^{(i)})$, *where* $Y_{k_i j_i}$ *are spherical harmonics in* $\mathbb{S}^{\zeta-1}$, *and* $N(\zeta, k_i)$ *are the number of harmonics of frequency* $k_i$ *in* $\mathbb{S}^{\zeta-1}$.

*Proof.* By the definition of the multivariate Gegenbauer polynomials and the univariate addition theorem [9] we get

$$Q_{\mathbf{k}}(\langle \mathbf{x}^{(1)}, \mathbf{z}^{(1)} \rangle, ..., \langle \mathbf{x}^{(i)}, \mathbf{z}^{(j)} \rangle, ..., \langle \mathbf{x}^{(d)}, \mathbf{z}^{(d)} \rangle) = Q_{k_1}(\langle \mathbf{x}^{(1)}, \mathbf{z}^{(1)} \rangle) \cdot \ldots \cdot Q_{k_d}(\langle \mathbf{x}^{(d)}, \mathbf{z}^{(d)} \rangle)$$

$$= \left( \frac{|\mathbb{S}^{\zeta-1}|}{N(\zeta, k_1)} \sum_{j_1=1}^{N(\zeta, k_1)} Y_{k_1,j_1}(\mathbf{x}^{(1)}) Y_{k_1,j_1}(\mathbf{z}^{(1)}) \right) \cdot \ldots \cdot \left( \frac{|\mathbb{S}^{\zeta-1}|}{N(\zeta, k_d)} \sum_{j_d=1}^{N(\zeta, k_d)} Y_{k_d,j_d}(\mathbf{x}^{(d)}) Y_{k_d,j_d}(\mathbf{z}^{(d)}) \right)$$

$$= \left( \prod_{i=1}^{d} \frac{|\mathbb{S}^{\zeta-1}|}{N(\zeta, k_i)} \right) \sum_{\mathbf{j}=(1,...,1)}^{\mathbf{j}=(N(\zeta, k_1),...,N(\zeta, k_d))} \prod_{i=1}^{d} Y_{k_i,j_i}(\mathbf{x}^{(i)}) Y_{k_i,j_i}(\mathbf{z}^{(i)})$$

$$:= \left( \prod_{i=1}^{d} \frac{|\mathbb{S}^{\zeta-1}|}{N(\zeta, k_i)} \right) \sum_{\mathbf{j}: j_i \in [N(\zeta, k_i)]} Y_{\mathbf{k},\mathbf{j}}(\mathbf{x}) Y_{\mathbf{k},\mathbf{j}}(\mathbf{z}).$$

Note that the homogeneity of the SH-products $Y_{\mathbf{k},\mathbf{j}}(\mathbf{x})$ is a direct result of the homogeneity of the spherical harmonics $Y_{k_i,j_i}$. $\square$

**Lemma A.5.** *The set* $\{Y_{\mathbf{k},\mathbf{j}}\}$ *are orthonormal w.r.t uniform measure in* $\mathbb{MS}(\zeta, d)$.

*Proof.* We have that

$$\int_{\mathbb{MS}(\zeta,d)} Y_{\mathbf{k},\mathbf{j}}(\mathbf{x})Y_{\mathbf{k}',\mathbf{j}'}(\mathbf{x})d\mathbf{x} = \int_{\mathbb{MS}(\zeta,d)} \left(\prod_{i=1}^{d} Y_{k_i,j_i}(\mathbf{x}^{(i)})\right)\left(\prod_{i=1}^{d} Y_{k_i',j_i'}(\mathbf{x}^{(i)})\right) d\mathbf{x}$$

$$= \prod_{i=1}^{d} \left(\int_{\mathbb{S}^{\zeta-1}} Y_{k_i,j_i}(\mathbf{x}^{(i)})Y_{k_i',j_i'}(\mathbf{x}^{(i)})d\mathbf{x}^{(i)}\right) = \prod_{i=1}^{d} \delta_{k_i,k_i'} \cdot \delta_{j_i,j_i'}.$$

$\square$

## A.2 Mercer's decomposition

In this section we prove that the eigenfunctions of multi dot-product kernels consist of products of spherical harmonics. We further provide a way to calculate the eigenvalues using products of Gegenbauer polynomials.

**Lemma A.6.** *Let $\bar{k}$ be a multi-dot product kernel. Then, the eigenfunctions of $\bar{k}(\mathbf{x},\cdot)$ w.r.t uniform measure on $\mathbb{MS}(\zeta,d)$ are the SH-products. Namely, the eigenfunctions are*

$$\left\{Y_{\mathbf{k},\mathbf{j}}(\mathbf{x}) = \prod_{i=1}^{d} Y_{k_i j_i}\left(\mathbf{x}^{(i)}\right)\right\}_{\mathbf{k}\geq 0,\ j_i \in [N(q,k_i)]},$$

*where $Y_{k_i j_i}$ are the Spherical Harmonics in $\mathbb{S}^{\zeta-1}$, and $N(\zeta,k_i)$ are the number of harmonics of frequency $k_i$ in $\mathbb{S}^{\zeta-1}$. The eigenvalues, $\lambda_{\mathbf{k}}$, can be calculated using products of (univariate) Gegenbauer polynomials as follows,*

$$\lambda_{\mathbf{k}} = C(\zeta,d) \int_{[-1,1]^d} \bar{k}(\mathbf{t}) \prod_{i=1}^{d} Q_{k_i}(t_i)(1-t_i^2)^{\frac{\zeta-3}{2}} d\mathbf{t}$$

*where $\{Q_k(t)\}$ is the set of orthogonal Gegenbauer polynomials w.r.t the weights $(1-t_i^2)^{\frac{\zeta-3}{2}}$, and $C(\zeta,d)$ is a constant that depends on both $\zeta$ and $d$.*

*Proof.* Let $\bar{k}$ be a multi-dot product kernel. By definition for such kernel, there exists a multivariate analytic function $\kappa$ such that $\bar{k}^{(L)}(\mathbf{x},\mathbf{z}) = \kappa(\langle\mathbf{x}^{(1)},\mathbf{z}^{(1)}\rangle,...,\langle\mathbf{x}^{(d)},\mathbf{z}^{(d)}\rangle)$. Using lemma A.3, $\{Q_{\mathbf{k}}\}$ form an orthogonal basis in $[-1,1]^d$. Therefore, it can be readily shown (similar to [9]) that, $\kappa$ can be written as

$$\kappa(t_1,..,t_d) := \kappa(\mathbf{t}) = \sum_{\mathbf{k}\geq 0}\left(\prod_{i=1}^{d} N(\zeta,k_i)\frac{|\mathbb{S}^{\zeta-2}|}{|\mathbb{S}^{\zeta-1}|}\right)Q_{\mathbf{k}}(\mathbf{t})\int_{[-1,1]^d}\kappa(\tilde{\mathbf{t}})Q_{\mathbf{k}}(\tilde{\mathbf{t}})\prod_{i=1}^{d}(1-\tilde{t}_i^2)^{\frac{\zeta-3}{2}}d\tilde{\mathbf{t}} := \sum_{\mathbf{k}\geq 0}\lambda_{\mathbf{k}}Q_{\mathbf{k}}(\mathbf{t}).$$

Lemma A.4 implies

$$Q_{\mathbf{k}}(\langle\mathbf{x}^{(1)},\mathbf{z}^{(1)}\rangle,..,\langle\mathbf{x}^{(i)},\mathbf{z}^{(j)}\rangle,..,\langle\mathbf{x}^{(d)},\mathbf{z}^{(d)}\rangle) = \frac{|\mathbb{S}^{\zeta-1}|^d}{\prod_{i=1}^{d}N(\zeta,k_i)}\sum_{\mathbf{j}:j_i\in[N(\zeta,k_i)]}Y_{\mathbf{k},\mathbf{j}}(\mathbf{x})Y_{\mathbf{k},\mathbf{j}}(\mathbf{z}),$$

yielding

$$\bar{k}(\mathbf{x},\mathbf{z}) = \sum_{\mathbf{k}\geq 0}\lambda_{\mathbf{k}}\sum_{\mathbf{j}:j_i\in[N(\zeta,k_i)]}Y_{\mathbf{k},\mathbf{j}}(\mathbf{x})Y_{\mathbf{k},\mathbf{j}}(\mathbf{z}).$$

Since $\{Y_{\mathbf{k},\mathbf{j}}(\mathbf{x})\}$ are orthonormal w.r.t. the uniform measure in $\mathbb{MS}(\zeta,d)$ (Lemma A.5) we obtain that $\{Y_{\mathbf{k},\mathbf{j}}(\mathbf{x})\}$ are the eigenfunctions of $\bar{k}^{(L)}$, with the corresponding eigenvalues $\{\lambda_{\mathbf{k}} = |\mathbb{S}^{\zeta-2}|^d\int_{[-1,1]^d}\bar{k}(\mathbf{t})\prod_{i=1}^{d}Q_{k_i}(t_i)(1-t_i^2)^{\frac{\zeta-3}{2}}d\mathbf{t}\}$. $\square$

## A.3 Proof of Lemma 3.1

**Lemma A.7.** *Let $\bar{k}$ be a multi-dot product kernel with the power series given in (2), where $\mathbf{x}^{(i)},\mathbf{z}^{(i)} \in \mathbb{S}^{\zeta-1}$ respectively are pixels in $\mathbf{x},\mathbf{z}$. Then, the eigenvalues $\lambda_{\mathbf{k}}(\bar{k})$ of $\bar{k}$ are given by $\lambda_{\mathbf{k}}(\bar{k}) =$*

$\left|\mathbb{S}^{\zeta-2}\right|^d \sum_{\mathbf{s}\geq 0} b_{\mathbf{k}+2\mathbf{s}} \prod_{i=1}^d \lambda_{k_i}(t^{k_i+2s_i})$, where $\left|\mathbb{S}^{\zeta-2}\right|$ is the surface area of $\mathbb{S}^{\zeta-2}$, and $\lambda_k(t^n)$ is the $k$'th eigenvalue of $t^n$, given by

$$\lambda_k(t^n) = \frac{n!}{(n-k)!2^{k+1}} \frac{\Gamma\left(\frac{\zeta-1}{2}\right)\Gamma\left(\frac{n-k+1}{2}\right)}{\Gamma\left(\frac{n-k+\zeta}{2}\right)}$$

if $n-k$ is even and non-negative, while $\lambda_k(t^n) = 0$ otherwise, and $\Gamma$ is the Gamma function.

*Proof.* The proof follows the linearity of the integral operator. Let

$$\mathbb{k}(\mathbf{x},\mathbf{z}) = \sum_{\mathbf{n}\geq 0} b_{\mathbf{n}} \langle \mathbf{x}^{(1)}, \mathbf{z}^{(1)}\rangle^{n_1} \cdot ... \cdot \langle \mathbf{x}^{(d)}, \mathbf{z}^{(d)}\rangle^{n_d}, \tag{1}$$

and denote by $C(\zeta,d) = \left|\mathbb{S}^{\zeta-2}\right|^d$. Following Lemma A.6 the eigenvalues of $\mathbb{k}$ are given by

$$\begin{aligned}
\lambda_{\mathbf{k}} =& C(\zeta,d)\int_{[-1,1]^d} \mathbb{k}(\mathbf{t})\prod_{i=1}^d Q_{k_i}(t_i)(1-t_i^2)^{\frac{\zeta-3}{2}}dt_1...dt_d \\
=& C(\zeta,d)\int_{[-1,1]^d} \sum_{\mathbf{n}\geq 0} b_{\mathbf{n}}\mathbf{t^n}\prod_{i=1}^d Q_{k_i}(t_i)(1-t_i^2)^{\frac{\zeta-3}{2}}dt_1...dt_d \\
=& C(\zeta,d)\sum_{\mathbf{n}\geq 0} b_{\mathbf{n}}\prod_{i=1}^d\left(\int_{[-1,1]} t_i^n Q_{k_i}(t_i)(1-t_i^2)^{\frac{\zeta-3}{2}}dt_i\right) = C(\zeta,d)\sum_{\mathbf{n}\geq 0} b_{\mathbf{n}}\prod_{i=1}^d \lambda_{k_i}(t^{n_i}).
\end{aligned}$$

Also note from [1] that $\lambda_k(t^n) = 0$ whenever $n-k$ is either odd or negative, implying the statement of the lemma. $\qquad\square$

A consequence of the lemma above is that the eigenvalues of a kernel $\mathbb{k}$ can be bounded by the eigenvalues of other kernels if the power series coefficients of $\mathbb{k}$ are bounded by the respective coefficients of the other kernels. We summarize this in the following corollary:

**Corollary A.8.** *Let $\mathbb{k}, \mathbb{k}^{upper}, \mathbb{k}^{lower} : \mathbb{MS}(\zeta,d) \to \mathbb{R}$ be multi-dot product kernels. Assuming that for $\mathbf{t}\in[-1,1]^d$,*

$$\mathbb{k}(\mathbf{t}) = \sum_{\mathbf{n}} b_{\mathbf{n}}\mathbf{t^n}$$

$$\mathbb{k}^{upper}(\mathbf{t}) = \sum_{\mathbf{n}} b_{\mathbf{n}}^{upper}\mathbf{t^n}$$

$$\mathbb{k}^{lower}(\mathbf{t}) = \sum_{\mathbf{n}} b_{\mathbf{n}}^{lower}\mathbf{t^n}$$

*and suppose there exists $\mathbf{k}_0$ such that for all $\mathbf{n}\geq\mathbf{k}_0$, $0\leq c_1 b_{\mathbf{n}}^{lower}\leq b_{\mathbf{n}}\leq c_2 b_{\mathbf{n}}^{upper}$, with $c_1, c_2 > 0$. Then, for all $\mathbf{k}\geq\mathbf{k}_0$,*

$$c_1\lambda_{\mathbf{k}}(\mathbb{k}^{lower})\leq\lambda_{\mathbf{k}}(\mathbb{k})\leq c_2\lambda_{\mathbf{k}}(\mathbb{k}^{upper}) \tag{2}$$

This corollary is an immediate result from Lemma 3.1.

# B Factorizable kernels

In this section we prove results presented in Section 3.2. We prove Theorem 3.2, which determines the eigenvalues of factorizable kernels whose power series coefficients decay at a polynomial rate. The following supporting lemma proves the theorem for $d=1$.

**Lemma B.1.** *Let $\tilde{\kappa}(t) = \sum_{n=0}^{\infty}\tilde{a}_n t^n$ where $\tilde{a}_n = O(n^{-\nu})$ with $\nu > 1$ and not integer. Then, the eigenvalues of $\tilde{\kappa}$ w.r.t. the uniform measure in $\mathbb{S}^{\zeta-1}$ are*

$$\lambda_k = \Theta\left(k^{-(\zeta+2\nu-3)}\right).$$

*Proof.* By applying Corollary A.8 with $d = 1$ we have that if $f(t) = \sum_{n=0}^{\infty} a_n t^n$ and $g(t) = \sum_{n=0}^{\infty} b_n t^n$ with $c_1 a_n \leq b_n \leq c_2 a_n$ then it holds that $\lambda_k(g) = \Theta(\lambda_k(f))$. It is therefore enough to find $f(t) = \sum_{n=0}^{\infty} \tilde{a}_n t^n$ where $\tilde{a}_n = O(n^{-\nu})$ and then calculate its eigenvalues. By [5] (Thm. VI.1, page 381), the function $f(t) = (1 - t)^{\nu-1}$, where $\nu > 1$ is non-integer, satisfies $f(t) = \sum_{n=0}^{\infty} \tilde{a}_n t^n$ with $\tilde{a}_n = O(n^{-\nu})$. Moreover, according to [2] (Thm. 7, page 17), the eigenvalues of $f(t) = (1 - t)^{\nu-1}$ in $\mathbb{S}^{\zeta-1}$ are

$$\lambda_k(f) = c_1 k^{-(\zeta+2\nu-3)},$$

which concludes the proof. $\qquad\square$

Relying on the lemma, we can now prove Theorem 3.2.

**Theorem B.2.** *Let $\mathbb{k}$ be a factorizable multi-dot product kernel, and let $\mathcal{R} \subseteq [d]$ denote its receptive field. Suppose that $\mathbb{k}$ can be written as a multivariate power series, $\mathbb{k}(\mathbf{t}) = \sum_{\mathbf{n} \geq 0} b_{\mathbf{n}} \mathbf{t}^{\mathbf{n}}$ with*

$$b_{\mathbf{n}} \sim c \prod_{i \in \mathcal{R},\, n_i > 0} n_i^{-\nu}.$$

*with constants $c > 0$, non-integer $\nu > 1$, and $b_{\mathbf{n}} = 0$ if $n_i > 0$ for any $i \notin \mathcal{R}$. Then the eigenfunctions of $\mathbb{k}$ w.r.t. the uniform measure are the SH-products, and its eigenvalues $\lambda_{\mathbf{k}}(\mathbb{k})$ satisfy*

$$\lambda_{\mathbf{k}} \sim \tilde{c} \prod_{i \in \mathcal{R},\, k_i > 0} k_i^{-(\zeta+2\nu-3)},$$

*where $\mathbf{k} \in \mathbb{N}^d$ be a vector of frequencies. Finally, $\lambda_{\mathbf{k}} = 0$ if $k_i > 0$ for any $i \notin \mathcal{R}$.*

*Proof.* Since $\mathbb{k}(\mathbf{t})$ is factorizable and can be written by a power series it can be written as

$$\mathbb{k}(\mathbf{t}) = c\tilde{\kappa}(t_1) \cdot ... \cdot \tilde{\kappa}(t_d),$$

where $\tilde{\kappa}(t) \sim \sum_{n=0}^{\infty} n^{-\nu} t^n$, and it can be readily shown that

$$\lambda_{\mathbf{k}}(\mathbb{k}) = c\lambda_{k_1}(\tilde{\kappa}) \cdot ... \cdot \lambda_{k_d}(\tilde{\kappa}).$$

Using Lemma B.1 we have that

$$c\lambda_{k_1}(\tilde{\kappa}) \cdot ... \cdot \lambda_{k_d}(\tilde{\kappa}) \sim \tilde{c} \prod_{i \in R,\, k_i > 0} k_i^{-(\zeta+2\nu-3)},$$

which concludes our proof. $\qquad\square$

# C  Positional bias of eigenvalues

We next prove results presented in Section 3.3. We next prove Theorem 3.4.

**Theorem C.1.** *Let $\mathbb{k}^{(L)}$ be hierarchical and factorizable of depth $L > 1$ with filter size $q$, so that $\mathbb{k}^{(L)}(\mathbf{t}) = \sum_{\mathbf{n} \geq 0} b_{\mathbf{n}} \mathbf{t}^{\mathbf{n}} = c \sum_{\mathbf{n} \geq 0} a_{n_1} \cdot ... \cdot a_{n_d} \mathbf{t}^{\mathbf{n}}$ with $a_0 > 0$ and $a_{n_i} = n_i^{-\nu}$ for $\nu > 1$. Then there exist a scalar $A = 1 + \frac{1}{a_0}$ such that:*

1. *The power series coefficients of $\mathbb{k}^{(L)}$ satisfy*

$$c_{A,\mathbf{n}} \mathbf{n}^{-\nu} \leq b_{\mathbf{n}},$$

   *where*

$$c_{A,\mathbf{n}} = c_L \prod_{i=1}^{d} A^{\min(p_i^{(L)}, n_i)}.$$

2. *The eigenvalues $\lambda_{\mathbf{k}}(\boldsymbol{k}^{(L)})$ satisfy*

$$c_{A,\mathbf{k}} \prod_{\substack{i=1 \\ n_i > 0}}^{d} k_i^{-(\zeta + 2\nu - 3)} \leq \lambda_{\mathbf{k}},$$

*where*

$$c_{A,\mathbf{k}} = \tilde{c}_L \prod_{i=1}^{d} A^{\min(p_i^{(L)}, k_i)}.$$

$c_L$ and $\tilde{c}_L$ are constants that depends on L, and $p_i^{(L)}$ denotes the number of paths from pixel $i$ to the output of $\boldsymbol{k}^{(L)}$.

To prove the theorem we provide several supporting lemmas and the following definition:

**Definition C.2.** A kernel $\tilde{\boldsymbol{k}}^{(L)}[-1,1]^{q^L} \to \mathbb{R}$ is called stride-q hierarchical of depth $L > 1$ if there exists a sequence of kernels $\tilde{\boldsymbol{k}}^{(1)}, ..., \tilde{\boldsymbol{k}}^{(L)}$ such that $\tilde{\boldsymbol{k}}^{(l)}(\mathbf{t}) = f\left(\tilde{\boldsymbol{k}}^{(l-1)}(\mathbf{t}_1), ..., \tilde{\boldsymbol{k}}^{(l-1)}(\mathbf{t}_q)\right)$ with $f : \mathbb{R}^q \to \mathbb{R}$, $\mathbf{t} = (\mathbf{t}_1, ..., \mathbf{t}_q) \in [-1,1]^{q^{l-1}}$ and $\boldsymbol{k}^{(1)}(t) = t \in [-1,1]$. A kernel $\boldsymbol{k}^{(L)} : [-1,1]^{q(L-1)+1} \to \mathbb{R}$ is stride-1 hierarchical if for all $1 < l \leq L$, $\boldsymbol{k}^{(l)} = f\left(\boldsymbol{k}^{(l-1)}(\mathbf{t}_1), \boldsymbol{k}^{(l-1)}(s_1\mathbf{t}_1), ..., \boldsymbol{k}^{(l-1)}(s_{q-1}\mathbf{t}_1)\right)$ and $\mathbf{t}_1 \in [-1,1]^{l(q-1)+1}$.

We next formulate the relation between the power series coefficient of the two kernels:

**Lemma C.3.** *Let $\boldsymbol{k}^{(L)}(\mathbf{t}) : [-1,1]^d \to \mathbb{R}$ be stride-1 kernel and $\tilde{\boldsymbol{k}}^{(L)}(\tilde{\mathbf{t}}) : [-1,1]^{q^L} \to \mathbb{R}$ be stride-q kernel. Then, there exists a variables substitution $S : [q^L] \to [d]$ such that if $\tilde{t}_{S(j)} = t_j$ for all $j \in [q^L]$ then*

$$\tilde{\boldsymbol{k}}^{(L)}(t_{S(0)}, .., t_{S(q^L-1)}) \equiv \boldsymbol{k}^{(L)}(t_0, .., t_{d-1}).$$

*Moreover, if $\boldsymbol{k}^{(L)}(\mathbf{t}) = \sum_{\mathbf{n} \geq 0} b_{\mathbf{n}} \mathbf{t}^{\mathbf{n}}$ and $\tilde{\boldsymbol{k}}^{(L)}(\tilde{\mathbf{t}}) = \sum_{\tilde{\mathbf{n}} \geq 0} \tilde{b}_{\tilde{\mathbf{n}}} \tilde{\mathbf{t}}^{\tilde{\mathbf{n}}}$ then*

$$b_{\mathbf{n}} = \sum_{\mathcal{S}} \tilde{b}_{\tilde{\mathbf{n}}}$$

*where $\mathcal{S} = \{\tilde{n}_0, .., \tilde{n}_{q^L-1} | \forall i = 0, .., d-1, \sum_{i=S(j)} \tilde{n}_j = n_i\}$.*

*Proof.* We assume here that $d \leq (q-1)L$, in any other case can take $mod(d)$. We construct the mapping $S$ and prove its correctness by induction. For any index $j = 0, .., q^L - 1$ we write $j = a_{L-1}q^{L-1} + a_{L-2}q^{L-2} + .. + a_1 q + a_0$ where $a_i = 0, 1, .., q-1$. Then, we define $S(j) := S_L(j) = a_{L-1} + a_{L-2} + .. + a_0$. We next prove by induction that $\tilde{\boldsymbol{k}}^{(L)}(t_{S(0)}, .., t_{S(q^{(L)}-1)}) \equiv \boldsymbol{k}^{(L)}(t_0, .., t_{d-1})$. For $L = 2$ we have:

$$\tilde{\boldsymbol{k}}^{(2)}(t_{S(0)}, .., t_{S(q^2-1)}) = f\left(\boldsymbol{k}^{(1)}(t_{S(0)}, .., t_{S(q-1)}), ..., \boldsymbol{k}^{(1)}(t_{S((q-1)q)}, .., t_{S(q^2-1)})\right)$$

$$= f\left(\boldsymbol{k}^{(1)}(t_0, .., t_{q-1}), ..., \boldsymbol{k}^{(1)}(t_{q-1}, .., t_{2q-2})\right) = f\left(\boldsymbol{k}^{(1)}(\mathbf{t}), \boldsymbol{k}^{(1)}(s_1 \mathbf{t}), ..., \boldsymbol{k}^{(1)}(s_{q-1} \mathbf{t})\right)$$

Where $\mathbf{t} = t_0, .., t_{q-1}$ and $s$ is the shift operator. This concludes the case of $L = 2$. For $L > 2$ we assume that $S_{L-1}(j) = a_{L-2} + .. + a_0$ is the correct assignment for $q^{L-1} - 1$ variables and get that

$$\tilde{\boldsymbol{k}}^{(L)}(t_{S(0)}, .., t_{S(q^L-1)}) = f\left(\tilde{\boldsymbol{k}}^{(L-1)}(t_{S(0)}, .., t_{S((q-1)q^{L-2}+..+q-1)}), ..., \tilde{\boldsymbol{k}}^{(L-1)}(t_{S((q-1)q^{L-1})}, .., t_{S(q^L-1)})\right)$$

$$= f\left(\tilde{\boldsymbol{k}}^{(L-1)}(t_0, .., t_{(L-1)(q-1)}), ..., \tilde{\boldsymbol{k}}^{(L-1)}(t_{(q-1)}, .., t_{L(q-1)})\right)$$

$$=^{(1)} f\left(\boldsymbol{k}^{(L-1)}(\mathbf{t}), ..., \boldsymbol{k}^{(L-1)}(s_{q-1} \mathbf{t})\right),$$

where $^{(1)}$ holds from the induction hypothesis and $\mathbf{t} = t_0, .., t_{(L-1)(q-1)}$. Therefore

$$\tilde{\boldsymbol{k}}^{(L)}(t_{S(0)}, .., t_{S(q^L-1)}) \equiv \boldsymbol{k}^{(L)}(t_0, .., t_{d-1}).$$

.

Finally since $f$ is an analytic function it holds that:

$$\boldsymbol{k}^{(L)}(\mathbf{t}) = \tilde{\boldsymbol{k}}^{(L)}(t_{S(0)}, .., t_{S(q^L-1)}) = \sum_{\tilde{\mathbf{n}} \geq 0} \tilde{b}_{\tilde{\mathbf{n}}} t_{S(0)}^{\tilde{n}_1} \cdots t_{S(q^L-1)}^{\tilde{n}_{q^L-1}} = \sum_{\mathbf{n} \geq 0} \mathbf{t}^{\mathbf{n}} \sum_{\mathcal{S}} \tilde{b}_{\tilde{\mathbf{n}}}$$

where $\mathcal{S} = \{\tilde{n}_0, .., \tilde{n}_{q^L-1} | \forall i = 0, .., d-1, \sum_{i=S(j)} \tilde{n}_j = n_i\}$. Therefore, from the uniqueness of the power series we get that

$$b_{\mathbf{n}} = \sum_{\mathcal{S}} \tilde{b}_{\mathbf{n}}.$$

$\square$

**Lemma C.4.** *Let* $\mathbf{k} \in \mathbb{N}^m$ *and consider the series* $S_m(n) = \sum_{k_1+...+k_m=n} \prod_{i=1}^{m} k_i^{-\nu} = \sum_{|\mathbf{k}|=n} \mathbf{k}^{-\nu}$ *with* $\nu > 1$ *and the convention* $0^{-\nu} = a_0 > 0$. *Then, for* $n \geq m$, $S_m(n)$ *is bounded from above and below as follows*

$$A^{m-1}n^{-\nu} \leq S_m(n) \leq B^{m-1}n^{-\nu}, \tag{3}$$

*with* $B > A = (a_0 + 1) > 1$ *constants.*

*Proof.* We show this by induction over $m$, i.e., the length of the vector $\mathbf{k}$. We begin by showing this for $S = S_2(n)$ for any $n \geq 2$, i.e., $An^{-\nu} \leq S = \sum_{k=0}^{\tilde{n}} k^{-\nu}(n-k)^{-\nu} \leq Bn^{-\nu}$ for constants $A$ and $B$.

**Lower bound.** For $n > 2$ it holds that

$$S = \sum_{k=0}^{n} k^{-\nu}(n-k)^{-\nu} = 2 \cdot a_0 \cdot n^{-\nu} + 2 \cdot (n-1)^{-\nu} + \sum_{k=2}^{n-2} k^{-\nu}(n-k)^{-\nu}$$

$$\geq 2 \cdot a_0 \cdot n^{-\nu} + 2 \cdot (n-1)^{-\nu}$$

$$\geq 2(a_0 + 1)n^{-\nu} \geq (2a_0 + 1)n^{-\nu}$$

$$\geq (a_0 + 1)n^{-\nu}.$$

For $n = 2$, we have that $S = 2a_0 n^{-\nu} + (n-1)^{-\nu} \geq (2a_0 + 1)n^{-\nu} \geq (a_0 + 1)n^{-\nu}$.

Therefore, it holds for $n \geq 2$ that $S_2(n) \geq A_n^{-\nu}$, where $A = a_0 + 1$.

**Upper bound.** We show that for $n \geq 2$ it holds that $n^{\nu}S_2(n) = n^{\nu}\sum_{k=0}^{n} k^{-\nu}(n-k)^{-\nu} \leq (2a_0 + 2) + \frac{2^{(\nu+1)}}{\nu-1}$. This follows from:

$$n^{\nu}\sum_{k=0}^{n} k^{-\nu}(n-k)^{-\nu} \leq (2a_0 + 2^{\nu+1}) + \sum_{k=2}^{n-2}\left(\frac{n-k+k}{k(n-k)}\right)^{\nu} = (2a_0 + 2^{\nu+1}) + \sum_{k=2}^{n-2}\left(\frac{n-k}{k(n-k)} + \frac{k}{k(n-k)}\right)^{\nu}$$

$$= (2a_0 + 2^{\nu+1}) + \sum_{k=2}^{n-2}\left(\frac{1}{k} + \frac{1}{(n-k)}\right)^{\nu} \leq (2a_0 + 2^{\nu+1}) + \sum_{k=2}^{n-2}\left(2\max\left\{\frac{1}{k}, \frac{1}{n-k}\right\}\right)^{\nu} \leq (2a_0 + 2^{\nu+1}) + 2^{\nu}2\sum_{k=2}^{n/2} k^{-\nu}.$$

Note that $f(k) = k^{-\nu}$ is monotonically decreasing and therefore can be bounded by the integral

$$\sum_{k=2}^{n/2} k^{-\nu} \leq \int_{1}^{n/2} \frac{1}{x^{\nu}}dx = \frac{1}{\nu-1} - \left(\frac{2}{n}\right)^{\nu-1}\frac{1}{\nu-1} \leq \frac{1}{\nu-1}$$

So overall we have that $n^{\nu}S_2(n) \leq 2a_0 + 2^{\nu+1} + \frac{2^{(\nu+1)}}{\nu-1}$ implying that $S_2(n) \leq Bn^{-\nu}$ with $B = 2a_0 + 2^{\nu+1} + \frac{2^{(\nu+1)}}{\nu-1}$.

**Induction step.** We next use induction to prove the lemma for $S_m(n)$ for $m > 2$ and $n \geq m$. Assume the lemma holds for $S_m$, i.e., $A^{m-1}n^{-\nu} \leq S_m(n) \leq B^{m-1}n^{-\nu}$ for $n \geq m$ and $A = a_0 + 1 > 1$, we aim to prove this for $S_{m+1}(n)$ for $n \geq m+1$.

$$S_{m+1}(n) = \sum_{k_1=0}^{n} k_1^{-\nu} \sum_{k_2+...+k_{m+1}=n-k_1} k_2^{-\nu} \cdots k_{m+1}^{-\nu}.$$

Using the induction assumption, we obtain

$$S_{m+1}(n) = \sum_{k_1=0}^{n} k_1^{-\nu} \sum_{k_2+...+k_{m+1}=n-k_1} k_2^{-\nu} \cdots k_{m+1}^{-\nu}$$

$$\geq a_0 \sum_{k_2+...+k_{m+1}=n} k_2^{-\nu} \cdots k_{m+1}^{-\nu} + \sum_{k_2+...+k_{m+1}=n-1} k_2^{-\nu} \cdots k_{m+1}^{-\nu}$$

$$\geq a_0 A^{m-1} n^{-\nu} + A^{m-1}(n-1)^{-\nu} \geq (a_0+1)^m n^{-\nu} = A^m n^{-\nu}.$$

Note that in the two sums above the induction assumption holds since $n \geq n-1 \geq m$. This concludes the proof for the lower bound. The proof for the upper bound proceeds in a similar way. $\square$

**Lemma C.5.** *Let* $\mathbf{k} \in \mathbb{N}^m$ *and consider the series* $S_m(n) = \sum_{|\mathbf{k}|=n} \mathbf{k}^{-\nu}$ *with* $\nu > 1$ *and the convention* $0^{-\nu} = a_0 > 0$. *Then, for* $2 \leq n \leq m$, $S_m(n)$ *is bounded from above and below as follows.*

$$a_0^{m-n} A^{n-1} n^{-\nu} \leq S_m(n) \leq a_0^{m-n} \left(\frac{m \cdot e}{n}\right)^n B^{n-1} n^{-\nu}, \tag{4}$$

*where* $A, B$ *are given in Lemma C.4. Note that for* $m = n$ *the lower bound boils down to the lower bound in Lemma C.4.*

*Proof.* We next prove the lemma for $2 \leq n \leq m$.

$$S_m(n) = \sum_{k_1+...+k_m=n} k_1^{-\nu} \cdot ... \cdot k_m^{-\nu} \geq a_0^{m-n} \sum_{k_1+...+k_n=n} k_1^{-\nu} \cdot ... \cdot k_n^{-\nu} \geq a_0^{m-n} A^{n-1} n^{-\nu},$$

where the last inequality holds from Lemma C.4 with $n = m$.

For the upper bound we have

$$S_m(n) = \sum_{k_1+..+k_m=n} k_1^{-\nu} \cdot ... \cdot k_m^{-\nu} \leq^{(1)} a_0^{m-n} \binom{m}{n} \sum_{k_1+..+k_n=n} k_1^{-\nu} \cdot ... \cdot k_n^{-\nu}$$

$$\leq^{(2)} a_0^{m-n} \binom{m}{n} B^{n-1} n^{-\nu} \leq a_0^{m-n} \left(\frac{m \cdot e}{n}\right)^n B^{n-1} n^{-\nu},$$

where $^{(1)}$ considers subsets of size $n$ and sets the remaining orders $k_i$ to zero. Note that since $n \leq m$ this covers all the options of satisfying the sum $k_1 + .. + k_m = n$ (with some repetitions). $^{(2)}$ uses the bound of Lemma C.4. $\square$

**Lemma C.6.** *Let* $\bar{k}^{(L)}$ *be an hierarchical factorizable kernel of depth $L$ and filter size $q$, where* $\bar{k}^{(L)}(\mathbf{t}) = \sum_{\mathbf{n} \geq 0} b_{\mathbf{n}} \mathbf{t}^{\mathbf{n}} = c \sum_{\mathbf{n} \geq 0} a_{n_1} \cdot ... \cdot a_{n_d} \mathbf{t}^{\mathbf{n}}$ *with $a_0 > 0$ and $a_{n_i} = n_i^{-\nu}$ for $\nu > 1$. Then, the Taylor coefficients of $\bar{k}^{(L)}$ satisfy*

$$c_{A,\mathbf{n}} \mathbf{n}^{-\nu} \leq b_{\mathbf{n}} \leq c_{B,\mathbf{n}} \mathbf{n}^{-\nu}$$

*where*

$$c_{A,\mathbf{n}} = c_L \prod_{i=1}^{d} \bar{A}^{\min(p_i^{(L)}, n_i)}$$

*and* $c_{B,\mathbf{n}} = \bar{c}_L \prod_{i=1}^{d} c_B(p_i^{(L)}, n_i)$

$$c_B(p_i^{(L)}, n_i) = \begin{cases} \left(\frac{p_i^{(L)} \cdot e}{n_i}\right)^{n_i} B^{n_i}, & 1 \leq n_i < p_i^{(L)} \\ B^{p_i^{(L)}}, & n_i \geq p_i^{(L)} \end{cases}$$

*with* $B \geq \bar{A} = 1 + \frac{1}{a_0}$ *and $c_L, \bar{c}_L$ are constants. $p_i^{(L)}$ denotes the number of paths from pixel $j$ to the output of the corresponding equivariant network.*

*Proof.* Since $\boldsymbol{k}^{(L)}$ is factorizable we can use the hierarchical stride $q$ kernel $\tilde{\boldsymbol{k}}^{(L)}(\tilde{\mathbf{t}})$ and write:

$$\tilde{\boldsymbol{k}}^{(L)}(\tilde{\mathbf{t}}) = \sum_{\tilde{\mathbf{n}} \geq 0} \tilde{b}_{\tilde{\mathbf{n}}} \mathbf{t}^{\tilde{\mathbf{n}}} = \sum_{\tilde{\mathbf{n}} \geq 0} a_{\tilde{n}_1} \cdot \ldots \cdot a_{\tilde{n}_{q^{L}-1}} \mathbf{t}^{\tilde{\mathbf{n}}}$$

with $a_{\tilde{n}_i} = \tilde{n}_i^{-\nu}$. Moreover using the mapping $S$ from lemma C.3 we have that $\boldsymbol{k}^{(L)}(\mathbf{t}) = \sum_{\mathbf{n} \geq 0} b_{\mathbf{n}} \mathbf{t}^{\mathbf{n}}$ with

$$b_{\mathbf{n}} = \sum_{S} \tilde{b}_{\tilde{\mathbf{n}}} = c \sum_{S} \tilde{\mathbf{n}}^{-\nu}$$

where $S = \{\tilde{n}_1, .., \tilde{n}_{q^{L}-1} | \forall i = 1, .., d, \sum_{i=S(j)} \tilde{n}_j = n_i\}$. Note that $|\{i | S(i) = j\}| = p_j^{(L)}$ where $p_j^{(L)}$ denotes the number of paths from the input pixel to the output, therefore by combining Lemma C.4 for the case of $p_j^{(L)} \geq n_j$ and Lemma C.5 for the case of $p_j^{(L)} \leq n_j$ we have that

$$\tilde{c}_{A,\mathbf{n}} \mathbf{n}^{-\nu} \leq b_{\mathbf{n}}$$

where $c_{A,\mathbf{n}} = \prod_{i=1}^{d} c(p_i^{(L)}, n_i)$ and

$$c(p_i^{(L)}, n_i) = \begin{cases} a_0^{p_i^{(L)} - n_i}(1 + a_0)^{n_i - 1}, & n_i < p_i^{(L)} \\ (1 + a_0)^{p_i^{(L)} - 1}, & n_i \geq p_i^{(L)} \end{cases}$$

So all in all we get

$$c(p_i^{(L)}, n_i) := (1 + a_0)^{-1} a_0^{p_i^{(L)}} A^{\min(p_i^{(L)}, n_i)}$$

with $A = 1 + \frac{1}{a_0}$. This leads to

$$c_{A,\mathbf{n}} = c_L \prod_{i=1}^{d} A^{\min(p_i^{(L)}, n_i)}$$

where $A = 1 + \frac{1}{a_0}$ and $c_L = (1 + a_0)^{-d} \cdot a_0^{\sum_{i=1}^{d} p_i^{(L)}}$. The same set of steps using lemmas C.4 and C.5 leads to the results of $c_{B,\mathbf{n}}$ $\qquad\square$

**Lemma C.7.** *Let $\boldsymbol{k}^{(L)}$ be a stride-1 hierarchical and factorizable of fixed depth $L$ and filter size $q$, where $\boldsymbol{k}^{(L)}(\mathbf{t}) = \sum_{\mathbf{n} \geq 0} b_{\mathbf{n}} \mathbf{t}^{\mathbf{n}} = c \sum_{\mathbf{n} \geq 0} a_{n_1} \cdot \ldots \cdot a_{n_d} \mathbf{t}^{\mathbf{n}}$ with $a_0 > 0$, and $a_{n_i} = n_i^{-\nu}$ for $\nu > 1$. Then, the eigenvalues $\lambda_{\mathbf{k}}$ of $\boldsymbol{k}^{(L)}$ satisfy*

$$\lambda_{\mathbf{k}} \geq c_{A,\mathbf{k}} \prod_{\substack{i=1 \\ n_i > 0}}^{d} k_i^{-(\zeta + 2\nu - 3)}$$

$$c_{A,\mathbf{k}} = c_L \prod_{i=1}^{d} A^{\min(p_i^{(L)}, k_i)},$$

*with $A = 1 + \frac{1}{a_0}$ and $p_i^{(L)}$ denotes the number of paths from pixel $i$ to the output of the corresponding equivariant network.*

*Proof.* Using Lemma C.6 we have

$$b_{\mathbf{n}} \geq c \prod_{i=1}^{d} A^{\min(p_i, n_i)} n_i^{-\nu}.$$

Using Lemma 3.1 we have

$$\lambda_{\mathbf{k}} = |\mathbb{S}^{\zeta - 2}|^d \sum_{\mathbf{s} \geq 0} b_{\mathbf{k}+2\mathbf{s}} \lambda_{\mathbf{k}}\left(\mathbf{t}^{\mathbf{k}+2\mathbf{s}}\right),$$

where we denote by $\lambda_{\mathbf{k}}\left(\mathbf{t}^{\mathbf{k}+2\mathbf{s}}\right)=\prod_{i=1}^{d}\lambda_{k_i}\left(t_i^{k_i+2s_i}\right)$. This implies that

$$\lambda_{\mathbf{k}} \geq c|\mathbb{S}^{\zeta-2}|^d \sum_{\mathbf{s}\geq 0}\prod_{i=1}^{d}A^{\min(p_i,k_i+2s_i)}(k_i+2s_i)^{-\nu}\lambda_{k_i}\left(t_i^{k_i+2s_i}\right).$$

Applying the distributive law

$$\lambda_{\mathbf{k}} \geq c|\mathbb{S}^{\zeta-2}|^d \prod_{i=1}^{d}\sum_{s_i\geq 0}A^{\min(p_i,k_i+2s_i)}(k_i+2s_i)^{-\nu}\lambda_{k_i}\left(t_i^{k_i+2s_i}\right)=\prod_{i=1}^{d}\lambda_{k_i}(\mathbf{k}_i),$$

where we define the kernel $\mathbf{k}_i(t)$ by the power series

$$\mathbf{k}_i(t) = \sum_{n_j=0}^{\infty}c^{1/d}A^{\min(p_i,n_j)}n_j^{-\nu}t^{n_j}.$$

Therefore,

$$\lambda_{\mathbf{k}} \geq c\prod_{i=1}^{d}\left(\sum_{n_i=0}^{\infty}A^{\min(p_i,n_i)}n_i^{-\nu}\lambda_{k_i}\left(t_i^{n_i}\right)\right) = c\prod_{i=1}^{d}\left(\sum_{s_i=0}^{\infty}A^{\min(p_i,k_i+2s_i)}(k_i+2s_i)^{-\nu}\lambda_{k_i}\left(t^{k_i+2s_i}\right)\right)$$

$$\geq c\prod_{i=1}^{d}A^{\min(p_i,k_i)}\left(\sum_{s_i=0}^{\infty}(k_i+2s_i)^{-\nu}\lambda_{k_i}\left(t^{k_i+2s_i}\right)\right).$$

Therefore, using Theorem 3.2 we get that

$$\lambda_{\mathbf{k}} \geq c_{A,\mathbf{k}}\prod_{\substack{i=1\\n_i>0}}^{d}k_i^{-(\zeta+2\nu-3)}$$

$$c_{A,\mathbf{k}} = c_L\prod_{i=1}^{d}A^{\min(p_i^{(L)},k_i)}.$$

$\square$

# D   Kernels associated with the equivariant network

In this section we prove Theorem 3.5 presented in Section 3.4.

**Theorem D.1.** *Let $\mathbf{k}^{(L)}$ denote either CGPK-EqNet or CNTK-EqNet of depth L whose input includes $\zeta$ channels, with receptive field $\mathcal{R}$ and with ReLU activation. Then,*

1. *$\mathbf{k}^{(L)}$ can be written as a power series, $\mathbf{k}^{(L)}(\mathbf{t})=\sum_{\mathbf{n}\geq 0}b_{\mathbf{n}}\mathbf{t}^{\mathbf{n}}$ with*

$$c_1\prod_{i\in\mathcal{R},n_i>0}n_i^{-\nu_a} \leq b_{\mathbf{n}} \leq c_2\prod_{i\in\mathcal{R},n_i>0}n_i^{-\nu_b},$$

2. *The eigenvalues of $\mathbf{k}^{(L)}$ are bounded by*

$$c_3\prod_{\substack{i\in\mathcal{R}\\k_i>0}}k_i^{-(\zeta+2\nu_a-3)} \leq \lambda_{\mathbf{k}} \leq c_4\prod_{\substack{i\in\mathcal{R}\\k_i>0}}k_i^{-(\zeta+2\nu_b-3)},$$

*where for CGPK-EqNet $\nu_a=2.5$ and $\nu_b=1+3/(2d)$, while for CNTK-EqNet $\nu_a=2.5$ and $\nu_b=1+1/(2d)$ and $c_1,c_2,c_3$ and $c_4$ depend on L.*

We begin by proving the lower bound for $b_{\mathbf{n}}$ of CGPK-EqNet.

**Lemma D.2.** *Let $\boldsymbol{k}^{(L)}$ be a CGPK-EqNet of depth L, filter size q with ReLU activation Then, $\boldsymbol{k}^{(L)}$ can be written as a power series, $\boldsymbol{k}^{(L)}(\mathbf{t}) = \sum_{\mathbf{n} \geq 0} b_{\mathbf{n}} \mathbf{t}^{\mathbf{n}}$ with*

$$c_1 \mathbf{n}^{-\nu} \leq b_{\mathbf{n}},$$

*where $c_1 > 0$ is constant if the receptive field of $\boldsymbol{k}^{(L)}$ includes $\mathbf{n}$ and zero otherwise and $\nu = 2.5$.*

*Proof.* We prove the lemma by induction on $L$. For $L = 1$

$$\boldsymbol{k}^{(1)}(\mathbf{t}) = \kappa_1(t_1) = \sum_{n=0}^{\infty} a_n t_1^n,$$

where the equality on the right provides the power series of $\kappa_1$. Consequently, for $\mathbf{n} = (n, 0, .., 0)$, $b_{\mathbf{n}} = a_n \sim n^{-\nu}$, and the receptive field contains only one pixel. Therefore, $c_1$ is constant if $\mathbf{n} = (n, 0, .., 0)$ and zero otherwise. For $L > 1$ we denote $\kappa_1(u) = \sum_{n=0}^{\infty} a_n u^n$ and $g(\mathbf{t}) = \boldsymbol{k}^{L-1}(\mathbf{t}) = \sum_{\mathbf{n} \geq 0} \tilde{b}_{\mathbf{n}} \mathbf{t}^{\mathbf{n}}$ with the induction assumption that $\tilde{b}_{\mathbf{n}} \geq c n^{-\nu}$. Then we have that

$$\boldsymbol{k}^{L}(\mathbf{t}) = \kappa_1 \left( \frac{1}{q} \sum_{j=0}^{q-1} g(s_j \mathbf{t}) \right) = \sum_{n=0}^{\infty} \frac{a_n}{q^n} \left( \sum_{j=0}^{q-1} g(s_j \mathbf{t}) \right)^n$$

$$= \sum_{n=0}^{\infty} \frac{a_n}{q^n} \sum_{|\mathbf{k}|=n} \binom{n}{\mathbf{k}} \prod_{i=0}^{q-1} g^{k_i}(s_i \mathbf{t}) =^{(1)} \sum_{\mathbf{k} \geq 0} \frac{a_{|\mathbf{k}|}}{q^{|\mathbf{k}|}} \binom{|\mathbf{k}|}{\mathbf{k}} \prod_{i=0}^{q-1} \left( \sum_{\mathbf{m} \geq 0} \tilde{b}_{s_{-i}\mathbf{m}} \mathbf{t}^{s_{-i}\mathbf{m}} \right)^{k_i} := \sum_{\mathbf{n} \geq 0} b_{\mathbf{n}} \mathbf{t}^{\mathbf{n}},$$

where $^{(1)}$ is due to the fact that $g(s_i \mathbf{t}) = \sum_{\mathbf{m} \geq 0} \tilde{b}_{\mathbf{m}} (s_i \mathbf{t})^{\mathbf{m}} = \sum_{\mathbf{m} \geq 0} \tilde{b}_{s_{-i}\mathbf{m}} \mathbf{t}^{s_{-i}\mathbf{m}}$. Next, using a multivariate version of the Faá di Bruno formula (see, e.g., [8]), we have that:

$$b_{\mathbf{n}} = \sum_{\mathbf{k} \geq 0} \frac{a_{|\mathbf{k}|}}{q^{|\mathbf{k}|}} \binom{|\mathbf{k}|}{\mathbf{k}} \sum_{\{\mathbf{n}_1,...,\mathbf{n}_q | \sum_{i=1}^{q} k_i \mathbf{n}_i = \mathbf{n}\}} \prod_{i=0}^{q-1} \hat{B}_{\mathbf{n}_i, k_i}(.., \tilde{b}_{s_{-i}\mathbf{m}}, ..), \tag{5}$$

where $\hat{B}_{\mathbf{n},k}(\cdot)$ denote ordinary multivariate Bell polynomials defined as

$$\hat{B}_{\mathbf{n},k}(x_{\mathbf{i}_1}, x_{\mathbf{i}_2}, ...) = \sum_{\bar{\mathcal{J}}_{\mathbf{n},k}} \frac{k!}{j_{\mathbf{i}_1}! j_{\mathbf{i}_2}! ...} x_{\mathbf{i}_1}^{j_{\mathbf{i}_1}} x_{\mathbf{i}_2}^{j_{\mathbf{i}_2}} ...$$

and $\bar{\mathcal{J}}_{\mathbf{n},k} = \{j_{\mathbf{i}_1} + j_{\mathbf{i}_2} + ... = k \in \mathbb{R}; j_{\mathbf{i}_1} \mathbf{i}_1 + j_{\mathbf{i}_2} \mathbf{i}_2 + ... = \mathbf{n} \in \mathbb{R}^d\}$. Since all terms in (5) are non-negative, it suffices to choose one term to get a lower bound. Specifically, we choose $\mathbf{k} = (1, 1.., 1) \in \mathbb{R}^q$ and $\mathbf{n}_1, \mathbf{n}_q$ such that $\mathbf{n}_1 + \mathbf{n}_q = \mathbf{n}$, $\mathbf{n}_1^T \mathbf{n}_q = 0$, and $\mathbf{n}_i = 0$ for $i \notin \{1, q\}$. Noting that $|\mathbf{k}| = q$, $\hat{B}_{\mathbf{n}_1, 1} = \tilde{b}_{\mathbf{n}_1}$ and $\hat{B}_{\mathbf{n}_q, 1} = \tilde{b}_{\mathbf{n}_q}$, and $\hat{B}_{\mathbf{0}, 1} = b_0$, we obtain

$$b_{\mathbf{n}} \geq \frac{a_q}{q^q} q! \tilde{b}_0^{q-2} \tilde{b}_{\mathbf{n}_1} \tilde{b}_{\mathbf{n}_q} = C_q \tilde{b}_{\mathbf{n}_1} \tilde{b}_{\mathbf{n}_q} \geq^{(1)} C_q c^2 \mathbf{n}^{-\nu},$$

where $C_q = \frac{q^q}{q!} a_q \tilde{b}_0^{q-2}$ and $^{(1)}$ is due to the induction hypothesis. $\qquad \square$

**Corollary D.3.** *The bound in Lemma D.2 holds also for CNTK-EqNet.*

*Proof.* Let $\boldsymbol{k}^{(L)}$ be a CNTK-EqNet. Denote by $b_{\mathbf{n}}(\boldsymbol{k}^{(L)})$ as the power series coefficients of $\boldsymbol{k}^{(L)}$. Then, by definition,

$$\Sigma_{i,j}^{(l)}(\mathbf{x}, \mathbf{z}) = \kappa_1 \left( \frac{1}{q} \sum_{r=0}^{q-1} \tilde{\Sigma}_{i+r,j+r}^{(l-1)}(\mathbf{x}, \mathbf{z}) \right)$$

$$\Theta_{i,j}^{(l)}(\mathbf{x}, \mathbf{z}) = \frac{1}{q} \sum_{r=0}^{q-1} \left[ \kappa_0 \left( \tilde{\Sigma}_{i+r,j+r}^{(l-1)}(\mathbf{x}, \mathbf{z}) \right) \tilde{\Theta}_{i+r,j+r}^{(l-1)}(\mathbf{x}, \mathbf{z}) + \tilde{\Sigma}_{i+r,j+r}^{(l)}(\mathbf{x}, \mathbf{z}) \right],$$

Since $\kappa_0$ and $\kappa_1$ have only positive power series coefficients it holds that $b_{\mathbf{n}}(\boldsymbol{k}^{(L)}) = b_{\mathbf{n}}(\Theta_{i,i}^{(L)}) \geq \frac{c_\sigma}{q} b_{\mathbf{n}}(\tilde{\Sigma}_{i,i}^{(L)})$. Note that $\tilde{\Sigma}_{i,i}^{(L)}$ is the CGPK-EqNet of $L$ layers and therefore we can apply the lower bound of Lemma D.2 to get $b_{\mathbf{n}}(\boldsymbol{k}^{(L)}) \geq \frac{c_\sigma}{q} c_1 \mathbf{n}^{-\nu}$. $\qquad \square$

Next we give a general upper bound. We will use the following lemma: To prove the above lemma we will use the following supporting lemma

**Lemma D.4.** *Let $\boldsymbol{k}^{(L)}(\mathbf{t})$ be either CGPK-EqNet or CNTK-EqNet of depth $L$ with filter size $q$. Let $K_L^{FC}(u)$ be a fully connected kernel (NTK or GPK receptively) of one variable $u$. Then, plugging $t_1 = t_2.. = t_i = u$ to $\boldsymbol{k}^{(L)}(\mathbf{t})$ gives that $\boldsymbol{k}^{(L)}(\mathbf{t}) = K_L^{FC}(u)$, where $K_L^{FC}(u)$ denotes the corresponding CGPK or CNTK kernel of depth $L$ for a fully connected network.*

*Proof.* We prove the lemma for CGPK. The proof for CNTK is similar. We perform induction on $L$. For $L = 1$ the claim is trivial. For $L > 1$ plugging $t_1 = ... = t_i = u$ to $\boldsymbol{k}^{(L)}(\mathbf{t})$ together with the induction hypothesis gives us

$$\boldsymbol{k}^{(L)}(\mathbf{t}) = \kappa_1 \left( \frac{c_\sigma}{q} \sum_{j=0}^{q-1} \boldsymbol{k}^{(L-1)}(s_j \mathbf{t}) \right)$$

$$= \kappa_1 \left( \frac{c_\sigma}{q} \sum_{j=0}^{q-1} K_{L-1}^{FC}(u) \right) = \kappa_1(K_{L-1}^{FC}(u)) = K_L^{FC}(u).$$

$\square$

**Lemma D.5.** *Let $\boldsymbol{k}^{(L)}$ be either CNTK-EqNet or CGPK-EqNet of depth $L$ with filter size $q$ and ReLU activation. Then, $\boldsymbol{k}^{(L)}$ can be written as a power series, $\boldsymbol{k}^{(L)}(\mathbf{t}) = \sum_{\mathbf{n} \geq 0} b_\mathbf{n} \mathbf{t}^\mathbf{n}$, with, $\sum_{|\mathbf{n}|=k} b_\mathbf{n} = \Theta(a_k)$ where $a_k = k^{-\nu}$ with $\nu = 2.5$ for CPGK and $\nu = 1.5$ for CNTK.*

*Proof.* Let $\kappa(t) = \sum_{n=0}^{\infty} a_n t^n$. Using results by [3] (Theorem 8) we have that $K_L^{FC}(t) = \sum_{n=0}^{\infty} \tilde{a}_n t^n$ where $K_L^{FC}(t)$ is the NTK or GPK model for a FC network and $\tilde{a}_n = \Theta(n^{-\nu})$ for $\nu = 2.5, \nu = 1.5$ for GPK and NTK respectively. Moreover, we have that

$$\boldsymbol{k}^{(L)}(\mathbf{t}) = \sum_{\mathbf{n} \geq 0} b_\mathbf{n} \mathbf{t}^\mathbf{n}.$$

This, together with Lemma D.4 and plugging $t_1 = t_2 = .. = t_l = u$, yields

$$\boldsymbol{k}^{(L)}(\mathbf{t}) = \sum_\mathbf{n} b_\mathbf{n} \mathbf{t}^\mathbf{n} = \sum_\mathbf{n} b_\mathbf{n} u^{|\mathbf{n}|} = \sum_{k=0}^{\infty} u^k \sum_{|\mathbf{n}|=k} b_\mathbf{n}.$$

The uniqueness of power series further implies

$$\sum_{|\mathbf{n}|=k} b_\mathbf{n} = \tilde{a}_k = \Theta(k^{-\nu}),$$

which concludes the proof. $\square$

Next we upper bound $b_\mathbf{n}$ (Lemma D.7). We begin with a simple supporting lemma

**Lemma D.6.** *For any $d \geq 1$ positive (even) numbers $c_1, .., c_d \geq 1$, denote the two set of indices*

$$I_1 = \{(i_1, .., i_d) \in \mathbb{N}_+ \times .. \times \mathbb{N}_+ | c_k/2 \leq i_k \leq c_k\}$$
$$I_2 = \{(i_1, .., i_d) \in \mathbb{N}_+ \times .. \times \mathbb{N}_+ | (i_1 + .. + i_d) \in [c_1/2 + .. + c_d/2, c_1 + .. + c_d]\}.$$

*Then $I_1 \subseteq I_2$.*

*Proof.* Let $(i_1, .., i_d) \in I_1$. Then,

$$c_1/2 + .. + c_d/2 \leq i_1 + .. + i_d \leq c_1 + .. + c_d,$$

implying that $(i_1, .., i_d) \in I_2$. $\square$

**Lemma D.7.** *Let $\boldsymbol{k}(\mathbf{t}) = \sum_{\mathbf{n} \geq 0} b_\mathbf{n} \mathbf{t}^\mathbf{n}$ such that $\sum_{|\mathbf{n}|=n} b_\mathbf{n} = a_n \sim n^{-\nu}$ with $\nu > 1$. Then, there exists $c > 0$ such that $b_\mathbf{n} \leq c\mathbf{n}^{-\left(\frac{\nu-1}{d}+1\right)}$. The implication for CNTK-EqNet ($\nu = 1.5$) and for CGPK-EqNet ($\nu = 2.5$) can appear in a separate lemma.*

*Proof.* Let $n_1, .., n_d \gg 1$ be large enough and denote by $\bar{n} = \sum_{j=1}^{d} n_j$. Denote by $a_k = c \cdot k^{-\nu}$. By Lemma D.5 we have that $\sum_{|\mathbf{n}|=k} b_{\mathbf{n}} \leq C a_k$ . Therefore,

$$\sum_{k=\bar{n}/2}^{\bar{n}} \left( \sum_{|\mathbf{n}|=k} b_{\mathbf{n}} \right) = \sum_{|\mathbf{n}|=\bar{n}/2}^{\bar{n}} b_{\mathbf{n}} \leq C \sum_{k=\bar{n}/2}^{\bar{n}} a_k.$$

Here we can estimate the RHS using an integral and get

$$\sum_{k=\bar{n}/2}^{\bar{n}} a_k \approx \int_{\bar{n}/2}^{\bar{n}} \frac{1}{x^\nu} dx = (\nu - 1)(2^{(\nu-1)} - 1)\bar{n}^{-(\nu-1)}.$$

on the other hand, by denoting

$$I_1 = \{\bar{\mathbf{n}} \in \mathbb{N}_+ \times .. \times \mathbb{N}_+ | n_j/2 \leq \bar{n}_j \leq n_j\}$$
$$I_2 = \{\bar{\mathbf{n}} \in \mathbb{N}_+ \times .. \times \mathbb{N}_+ | |\bar{\mathbf{n}}| \in [n_1/2 + .. + n_d/2, n_1 + .. + n_d]\},$$

by Lemma D.6 and because $b_{\mathbf{n}} \geq 0$ we have that

$$\sum_{\mathbf{n} \in I_1} b_{\mathbf{n}} \leq \sum_{\mathbf{n} \in I_2} b_{\mathbf{n}} = \sum_{|\mathbf{n}|=\bar{n}/2}^{|\mathbf{n}|=\bar{n}} b_{\mathbf{n}} \leq C \sum_{k=\bar{n}/2}^{k=\bar{n}} a_k$$

Moreover $|I_1| = \frac{1}{2^d} n_1 \cdot ... \cdot n_d \cdot$ and the smallest element in the sum is $\min_{\mathbf{n} \in I_1}\{b_{\mathbf{n}}\} = b_{n_1,..,n_d}$. Therefore,

$$\frac{1}{2^d} n_1 \cdot ... \cdot n_d b_{n_1,..,n_d} \leq \sum_{\mathbf{n} \in I_1} b_{\mathbf{n}} \leq (\nu - 1)(2^{(\nu-1)} - 1)\bar{n}^{-(\nu-1)},$$

implying that

$$b_{n_1,..,n_d} \leq \frac{(\nu - 1)(2^{(\nu-1)} - 1)(n_1 + .. + n_d)^{-(\nu-1)}}{\frac{1}{2^d}(n_1 \cdot ... \cdot n_d)}.$$

Now applying the inequality of means we obtain $(n_1 + .. + n_d)/d \geq (n_1 \cdot ... \cdot n_d)^{\frac{1}{d}}$, and we finally get that

$$b_{n_1,..,n_d} \leq d 2^d (\nu - 1)(2^{(\nu-1)} - 1)(n_1 \cdot ... \cdot n_d)^{-\left(\frac{\nu-1}{d}+1\right)}.$$

$\square$

# E    Trace and GAP kernels

In this section we prove results presented in Section 3.5. We prove Theorem 3.7.

**Theorem E.1.** *Let $\bar{k}$ be a multi-dot-product kernel with Mercer's decomposition as in* (1)*, and let $k^{\mathrm{Tr}}$ and $k^{\mathrm{GAP}}$ respectively be its trace and GAP versions. Then,*

*1. $k^{\mathrm{Tr}}(\mathbf{x}, \mathbf{z}) = \sum_{\mathbf{k},\mathbf{j}} \lambda_{\mathbf{k}}^{\mathrm{Tr}} Y_{\mathbf{k},\mathbf{j}}(\mathbf{x}) Y_{\mathbf{k},\mathbf{j}}(\mathbf{z})$ with*

$$\lambda_{\mathbf{k}}^{\mathrm{Tr}} = \frac{1}{d} \sum_{i=0}^{d-1} \lambda_{s_i \mathbf{k}} \tag{6}$$

*Where $\lambda_{\mathbf{k}}$ denote the eigenvalues of $\bar{k}$.*

*2. $k^{\mathrm{GAP}}(\mathbf{x}, \mathbf{z}) = \sum_{\mathbf{k},\mathbf{j}} \lambda_{\mathbf{k}}^{\mathrm{Tr}} \tilde{Y}_{\mathbf{k},\mathbf{j}}(\mathbf{x}) \tilde{Y}_{\mathbf{k},\mathbf{j}}(\mathbf{z})$ with*

$$\tilde{Y}_{\mathbf{k},\mathbf{j}}(\mathbf{x}) = \frac{1}{\sqrt{d}} \sum_{i=0}^{d-1} Y_{s_i \mathbf{k}, s_i \mathbf{j}}(\mathbf{x}).$$

*Proof.* (1) Let $k^{\mathrm{Tr}}(\mathbf{x}, \mathbf{z})$ be a trace kernel. By definition

$$k^{\mathrm{Tr}}(\mathbf{x}, \mathbf{z}) = \frac{1}{d} \sum_{i=0}^{d-1} \mathring{k}(s_i \mathbf{x}, s_i \mathbf{z}),$$

where $\mathring{k}$ is a multi-dot-product kernel, with Mercer's decomposition

$$\mathring{k}(\mathbf{x}, \mathbf{z}) = \sum_{\mathbf{k},\mathbf{j}} \lambda_{\mathbf{k}} Y_{\mathbf{k},\mathbf{j}}(\mathbf{x}) Y_{\mathbf{k},\mathbf{j}}(\mathbf{z}).$$

Note that $\mathring{k}(s_i \mathbf{x}, s_i \mathbf{z})$ has the same eigenfunctions as $\mathring{k}(\mathbf{x}, \mathbf{z})$ with eigenvalues $\lambda_{s_i \mathbf{k}}$. So we get

$$k^{\mathrm{Tr}}(\mathbf{x}, \mathbf{z}) = \frac{1}{d} \sum_{i=0}^{d-1} \mathring{k}(s_i \mathbf{x}, s_i \mathbf{z}) = \frac{1}{d} \sum_{i=0}^{d-1} \sum_{\mathbf{k},\mathbf{j}} \lambda_{s_i \mathbf{k}} Y_{\mathbf{k},\mathbf{j}}(\mathbf{x}) Y_{\mathbf{k},\mathbf{j}}(\mathbf{z})$$

$$= \sum_{\mathbf{k},\mathbf{j}} \frac{1}{d} \sum_{i=0}^{d-1} \lambda_{s_i \mathbf{k}} Y_{\mathbf{k},\mathbf{j}}(\mathbf{x}) Y_{\mathbf{k},\mathbf{j}}(\mathbf{z}) = \sum_{\mathbf{k},\mathbf{j}} Y_{\mathbf{k},\mathbf{j}}(\mathbf{x}) Y_{\mathbf{k},\mathbf{j}}(\mathbf{z}) \frac{1}{d} \sum_{i=0}^{d-1} \lambda_{s_i \mathbf{k}}.$$

Therefore, we have

$$\lambda_{\mathbf{k}}^{\mathrm{Tr}} = \frac{1}{d} \sum_{i=0}^{d-1} \lambda_{s_i \mathbf{k}}.$$

(2) Let $k^{\mathrm{GAP}}(\mathbf{x}, \mathbf{z})$ be GAP kernel. By definition we have that

$$k^{\mathrm{GAP}}(\mathbf{x}, \mathbf{z}) = \frac{1}{d^2} \sum_{i=0}^{d-1} \sum_{j=0}^{d-1} \mathring{k}(s_i \mathbf{x}, s_j \mathbf{z})$$

Where $\mathring{k}$ is a multi-dot-product kernel. Using Mercer's decomposition (1), we have

$$\mathring{k}(s_i \mathbf{x}, s_j \mathbf{z}) = \sum_{\mathbf{k},\mathbf{j}} \lambda_{\mathbf{k}} Y_{\mathbf{k},\mathbf{j}}(s_i \mathbf{x}) Y_{\mathbf{k},\mathbf{j}}(s_j \mathbf{z}) = \sum_{\mathbf{k},\mathbf{j}} \lambda_{\mathbf{k}} Y_{s_{-i}\mathbf{k}, s_{-i}\mathbf{j}}(\mathbf{x}) Y_{s_{-j}\mathbf{k}, s_{-j}\mathbf{j}}(\mathbf{z}).$$

Therefore,

$$k^{\mathrm{GAP}}(\mathbf{x}, \mathbf{z}) = \frac{1}{d^2} \sum_{i=0}^{d-1} \sum_{j=0}^{d-1} \mathring{k}(s_i \mathbf{x}, s_j \mathbf{z}) = \frac{1}{d^2} \sum_{i=0}^{d-1} \sum_{j=0}^{d-1} \sum_{\mathbf{k},\mathbf{j}} \lambda_{\mathbf{k}} Y_{s_{-i}\mathbf{k}, s_{-i}\mathbf{j}}(\mathbf{x}) Y_{s_{-j}\mathbf{k}, s_{-j}\mathbf{j}}(\mathbf{z})$$

$$= \sum_{\mathbf{k},\mathbf{j}} \frac{1}{d^2} \lambda_{\mathbf{k}} \sum_{i=0}^{d-1} \sum_{j=0}^{d-1} Y_{s_{-i}\mathbf{k}, s_{-i}\mathbf{j}}(\mathbf{x}) Y_{s_{-j}\mathbf{k}, s_{-j}\mathbf{j}}(\mathbf{z})$$

$$= \sum_{\mathbf{k},\mathbf{j}} \frac{1}{d^2} \lambda_{\mathbf{k}} \left( \sum_{i=0}^{d-1} Y_{s_{-i}\mathbf{k}, s_{-i}\mathbf{j}}(\mathbf{x}) \right) \left( \sum_{j=0}^{d-1} Y_{s_{-j}\mathbf{k}, s_{-j}\mathbf{j}}(\mathbf{z}) \right).$$

We can denote $\tilde{Y}_{\mathbf{k},\mathbf{j}}(\mathbf{x}) = \frac{1}{\sqrt{d}} \sum_{i=0}^{d-1} Y_{s_i \mathbf{k}, s_i \mathbf{j}}(\mathbf{x})$. Note that $\tilde{Y}_{\mathbf{k},\mathbf{j}}(\mathbf{x})$ is invariant to all circular shifts of indices. So we further denote by $\mathbf{k}/S$ the set of indices $\mathbf{k}$ modulu the set of circular shifts $s_0, s_1, .., s_{d-1}$ and write the last expression as

$$k^{\mathrm{GAP}}(\mathbf{x}, \mathbf{z}) = \sum_{\mathbf{k}} \sum_{\mathbf{j}} \frac{1}{d} \lambda_{\mathbf{k}} \tilde{Y}_{\mathbf{k},\mathbf{j}}(\mathbf{x}) \tilde{Y}_{\mathbf{k},\mathbf{j}}(\mathbf{z}) = \sum_{\mathbf{k}/S} \sum_{\mathbf{j}/S} \left( \frac{1}{d} \sum_{i=0}^{d-1} \lambda_{s_i \mathbf{k}} \right) \tilde{Y}_{\mathbf{k},\mathbf{j}}(\mathbf{x}) \tilde{Y}_{\mathbf{k},\mathbf{j}}(\mathbf{z})$$

$$= \sum_{\mathbf{k}/S} \sum_{\mathbf{j}/S} \lambda_{\mathbf{k}}^{\mathrm{Tr}} \tilde{Y}_{\mathbf{k},\mathbf{j}}(\mathbf{x}) \tilde{Y}_{\mathbf{k},\mathbf{j}}(\mathbf{z}).$$

We conclude that the eigenfunctions are $\tilde{Y}_{\mathbf{k},\mathbf{j}}(\mathbf{x})$, and the eigenvalues are the same as $\lambda^{\mathrm{Tr}}$. Moreover, note that for any $\mathbf{k}, \mathbf{k}'$ such that $\forall i, \ \mathbf{k} \neq s_i \mathbf{k}'$ it holds that $\forall i \ Y_{\mathbf{k},\mathbf{j}}(\mathbf{x}) \perp Y_{s_i \mathbf{k}', \mathbf{j}}(\mathbf{x})$. Therefore, $\tilde{Y}_{\mathbf{k},\mathbf{j}}(\mathbf{x}) \perp \tilde{Y}_{\mathbf{k}',\mathbf{j}}(\mathbf{x})$, implying that $\{\tilde{Y}_{\mathbf{k},\mathbf{j}}(\mathbf{x})\}$ form an orthonormal basis. $\qquad \square$