# OpenReview forum: "On the Spectral Bias of Convolutional Neural Tangent and Gaussian Process Kernels"
_NeurIPS.cc/2022/Conference — NeurIPS 2022 Accept_

### Official Review · Reviewer_KKbC · 2022-07-08

**Rating:** 7
**Confidence:** 4
**Soundness:** 2 fair
**Presentation:** 3 good
**Contribution:** 2 fair

**Summary:**

This paper studies over-parameters CNNs from the perspective of NTK and Gaussian process kernels. A concise analysis of the eigenvalues and eigenfunctions of these kernels are derived. In particular, bounds of the eigenvalues and localization properties of the eigenfunctions are given. They provide a better understanding of the inductive bias in CNNs.

**Questions:**

In the introduction, when you talk about inductive bias and spectral bias, do you mean the same thing (line 33-35)?

As mentioned in the introduction, the eigenvalues are larger for functions that are localized in the center of the receptive field? Are you talking about the invariant kernel case as well (defined from f^GAP)? According to Corollary 3.8, it seems so. It seems to me strange that this could be true as the function f^GAP(x,theta) in invariant to the (cyclic) translation of x. Could you explain more in the text line 280? Also is k_{i+j} defined for i+j <=d in Corollary 3.8?

The function f^Eq is only keeping the first pixel in the output layer, am I right? It is thus not really equivariant if I understood, and this is why you got the previous localized eigenfunctions, am I right?

Could you be more precise on what it means ~ in Theorem 3.2? Does that mean for k-> infty?

What is SH product in line 226?


**Ethics Review Area:**

["I don’t know"]

**Strengths And Weaknesses:**

The strength of this paper is the derivation of the eigenvalue bounds for various kernels with different translation invariant properties (equivariant, non-invariant, invariant), by exploring multi-dot product structures in these kernels. What I find is not very clear is about the positional bias of eigenvalues in Section 3.3 (see questions next). Also the role of the hierarchy, as mentioned in the abstract, is not very clear. More discussions about them are welcome.

---

> ### Author Response · Authors · 2022-08-02
> **Response to Reviewer KKbC**
>
> Thank you for the constructive remarks.
>
> *Positional bias of eigenvalues*: We apologize for this misunderstanding. Throughout the paper (including in Corollary 3.8) $p_i$ denotes the number of paths from input pixel $i$ to $f^{Eq}$. Consequently, in Theorem 3.5, the eigenvalues of the equivariant kernels are larger for functions localized in the center of the receptive field. In Corollary 3.8 (following Theorem 3.7) the eigenvalues of the trace and gap kernels are obtained by averaging the eigenvalues of the equivariant kernel for all possible shifts. The obtained averages are therefore invariant to shifts. Consequently, the eigenvalues of these kernels are larger for functions in which high frequencies are concentrated in a local neighbourhood (anywhere in the image) compared to functions in which the high frequencies are scattered in distant pixels. This is demonstrated in Figure 2(right) for eigenfunctions involving two pixels. We hope this also clarifies the text in line 280. Finally, the pixel indices are cyclic, so in Colloraly 3.8 when $i+j>d$ it should be replaced by $i+j-d$. We will clarify this in the paper.
>
> *Role of hierarchy*: Our analysis indicates that the buildup of hierarchical features in a deep convolutional network affects its inductive bias by affecting the leading multiplicative coefficient of the eigenvalues of the respective kernels (as is derived in Sec 3.3 and indicated in the lower bounds in Theorem 3.5 and Corollary 3.8). In particular, this leading coefficient indicates *quantitatively* which target functions are preferred by CGPK and CNTK. The magnitude of the leading coefficient is determined by: (1) the number of paths from an input to the equivariant output, $p_i$. This is determined by the Irwin–Hall distribution (line 198) as a function of the number of layers $L$ and the size of the convolution filter $q$; and (2) the pixels that participate in each of the spherical harmonic products. For example, Figure 2(right) shows for the trace kernel that the leading coefficient of eigenvalues corresponding to SH-products of two pixels decays exponentially with the distance between the two pixels. The speed of this decay is determined by $L$ and $q$ through the Irwin–Hall distribution and can be thought of as quantifying the “effective” receptive field of the network. Similar calculations can be applied to SH-products that involve more than two pixels.
>
> *Inductive bias vs. spectral bias*: In this paper, by inductive bias we refer to those functions that networks can or are more likely to learn. The analogy with NTK reveals that with training data distributed uniformly, fully connected networks are biased toward learning low frequency functions. We refer to this specific inductive bias as spectral bias. In lines 33-35, we intended to say that a derivation of the inductive bias of CNNs, particularly as implied by the spectral decomposition of their respective CNTK and CGPK, is still missing. We will clarify these terms in the paper.
>
> *The definition of $f^{Eq}$*: The function $f^{Eq}$ indeed only keeps the first pixel in the output layer. Indeed this is why we get the localized eigenfunctions. Note that, as we explain in lines 79-80, the tuple $(f^{Eq}({\bf x}),f^{Eq}(s_1{\bf x}),...,f^{Eq}(s_{d-1}{\bf x}))$ obtained by applying $f^{Eq}$ to all shifts of ${\bf x}$ results in an equivariant mapping, hence its name.
>
> *The symbol $\sim$ in Theorem 3.2*: The symbol indeed implies equality up to scale for $k \rightarrow \infty$ (as we define in lines 69-70).
>
> *SH product in line 226*: In this line we consider using as target function a product of spherical harmonics (SH), $Y_{k_1,j_1}(x^{(1)})\cdot…\cdot Y_{k_d,j_d}(x^{(d)})$, where in each factor a SH is applied to the channels of a different pixel. Specifically, here we consider the case that, say, $k_1=k$ varies and $k_i$ are constant ($2 \le i \le d$). Since this target function is an eigenfunction of the CGPK/CNTK, denote the corresponding eigenvalue by $\lambda_k$, then using the lower bound in Theorem 3.5 and relying on [6], learning this eigenfunction should take $O(1/\lambda_k)=O(k^{\zeta+2})$ gradient descent iterations.
>
> We hope with this response the paper is clearer and that subsequently you will consider raising your score. We are happy to provide more clarifications if needed.

---

> > ### Comment · Reviewer_KKbC · 2022-08-03
> > **one more question**
> >
> > Thanks for your reply and I'll be happy to raise my score to accept the paper.
> >
> > Regarding your reply about the notation ~, I have a followup question:it seems to me Theorem 3.2 is the only asymptotic result which shows the behavoir of lambda_k when |k| is large. How about the rest of the results (e.g. Theorem 3.4, 3.5),
> > are they valid for all k or only for large |k|? By reading the text, it seems to be valid for all k, then a discussion about the tightness
> > as an open question is welcome in the conclusion.

---

> > > ### Author Response · Authors · 2022-08-05
> > > **Authors' response**
> > >
> > > Thank you for your reconsideration. The inequalities in Theorems 3.4 and 3.5 hold up to global scale factors which are stated in the theorem, and so effectively they are asymptotic in |k|. We will add a discussion about this and the tightness as an open question in the conclusion.

---

> > > > ### Comment · Reviewer_KKbC · 2022-08-09
> > > > **clarification**
> > > >
> > > > If I understand you reply, the results in Theorem 3.4 and 3.5 only hold for large |k| and not for all k, isn't it?

---

> > > > > ### Author Response · Authors · 2022-08-09
> > > > > **clarification**
> > > > >
> > > > > The theorems with the appropriate global scale factors ($c_L,\tilde c_L,c_1,c_2,c_3,c_4$) are valid for all $k$. However, when restricted to larger $|k|$, due to these scale factors, the bounds become tighter.

---

### Official Review · Reviewer_Q2dg · 2022-07-11

**Rating:** 5
**Confidence:** 3
**Soundness:** 3 good
**Presentation:** 3 good
**Contribution:** 3 good

**Summary:**

The paper studies the spectral bias of convolutional neural networks using the neural tangent kernel (NTK) and the Gaussian process kernel. Specifically, the paper derives the eigenvalues and the speed of learning of the eigenfunctions and illustrates how this differs from the fully-connected networks that were studied previously. The kernels studied are not dot product kernels, e.g. as in the fully-connected network analysis.

**Questions:**

Is there a typo in line 77 with the inequality?

**Limitations:**

There are no limitations explicitly written in the paper.

**Strengths And Weaknesses:**

[+] Understanding the inductive bias of neural networks is an important topic and convolutional networks are definitely a critical component.

[+] The paper is well-written overall, offering insights on the spectral bias and the analysis required to achieve this using NTK.

[-] Despite the attempt to move towards practical networks, the current format of the paper ignores skip connections (that are widely used in most of the practical networks) and also operations such as normalization. The NTK analysis for residual networks (in the fully-connected case) has already been conducted, so it would be a more complete result to have such cases with residual skip as well.

[-] Even though there is a thorough theoretical analysis, the results illustrate that the convolutional networks learn higher frequency functions more efficiently than fully-connected ones (under some cases). However, there is weak empirical validation of this phenomenon (only Fig. 3). A stronger empirical evidence would make the claims of this work much stronger.

[-] Minor: The paper requires proof-reading, since there are few errors, especially with parenthesis. For instance, in line 80 there is a parenthesis closing without one opening before.

Overall, the paper makes a contribution towards understanding the inductive bias of neural networks, which is an important topic. However, there are few structural limitations at the moment, e.g. lack of empirical evidence, or lack of skip connections that are used in practice.

---

> ### Author Response · Authors · 2022-08-02
> **Response to Reviewer Q2dg**
>
> Thank you for the detailed and constructive remarks.
>
> *Skip connections and normalization*: Extending our analysis to residual networks is nontrivial due to the complicated structure of NTK for convolutional networks. We plan to address this question in future work. Jacot et al. (arXiv:1907.05715) showed that the common approach for layer normalization, which is applied in each layer before the ReLU activation, does not change the NTK, and so our analysis is relevant to this operation.
>
> *Weak empirical validation*: To address this remark, we trained two networks, a fully connected (4 layers, width 512) and a convolutional one (4 layers, 1000 channels), on target functions composed of SH-products with $d=8$ and $\zeta=2$. Each target function includes two either adjacent or distant pixels with the same frequency $k$ ($1 \le k \le 3$). The figure in https://github.com/submission5738/Graphs/blob/main/CNN%20vs%20FC.jpg shows indeed that, consistent with our analysis, the FC network converges more slowly than the CNN on all tested target functions. We are pleased to see this reassuring evidence.
>
> Thank you for the additional minor comments.

---

> > ### Comment · Reviewer_Q2dg · 2022-08-03
> > **Response to the authors**
> >
> > I am thankful to the authors for responding to our feedback and adding the synthetic experiment.
> >
> > The synthetic experiment added indeed verifies the theoretical intuition. Have the authors evaluated the spectral bias of CNNs used in practice? There is an increasing number of works that have used convolutions for spectral bias in practice. It would strengthen the submission even further to perform such an empirical analysis.
> >
> > Could the authors be more specific about the part that including the residual networks would be nontrivial in their case?
> >
> > In addition, even though few typos were reported, the manuscript has not been updated so far.

---

> > > ### Author Response · Authors · 2022-08-05
> > > **Authors' response**
> > >
> > > Thank you for these additional comments.
> > >
> > > *Empirical analysis*: This is an interesting question for future work. We plan to design realistic data sets in which we can predict the complexity of learning, and evaluate real architectures.
> > >
> > > *Residual networks*: Including residual networks requires resolving two new non-trivial issues: (1) A formula for CNTK for residual networks; In particular, the recursive formula of Res-NTK for fully connected networks involves recursion in opposite directions. These have non-trivial consequences on the receptive fields in the convolutional case. (2) A power series decay for the Res-NTK formula (required for utilizing Lemma D.5 in our supplementary material); this requires applications of some complex tools.
> > >
> > > We are currently working on revising the paper according to the comments given by all of the reviewers.

---

> > > > ### Author Response · Authors · 2022-08-09
> > > > **Experiments with real CNNs**
> > > >
> > > > Following your remark, we further ran experiments with ‘Deep image prior’ (DIP) [Ulyanov et al., CVPR 2018]. The results of these experiments indeed confirm that real convolutional networks exhibit the properties predicted by our work. In particular, (1) CNNs learn low-frequency target functions faster than high-frequency ones, and (2) real equivariant CNNs exhibit positional bias.
> > > >
> > > > *Experiment 1: frequency bias*. DIP is an equivariant deep generator that learns a mapping from random patches to the RGB color of a target image. In this experiment, we manipulate the input to DIP to control the frequency of the target function learned by DIP. We construct the inputs by composing two copies of the same pattern, where we add noise to one of the copies. Specifically, denote the size of the target image by $W \times H$. We construct a pattern $P$ of size $W \times H/2$ by randomly sampling each entry from ${\cal N}(0,1)$. For our final input we concatenate $P$ with $P+n$ where $n \sim {\cal N}(0,v_n)$, producing an input of size $W \times H$. Assuming a small variance $v_n$, DIP will need to learn a mapping from pairs of nearly identical patches (in corresponding positions in $P$ and $P+n$) to (generally) two different target color values, corresponding to colors in the top and bottom halves of the image. The smaller $v_n$ is, the higher is the frequency of the target function (since with a smaller $v_n$, the same change in RGB values occurs for a smaller change in the input). Our prediction, therefore, is that DIP will converge more slowly when $v_n$ is small than when $v_n$ is large. This indeed can be seen in the plot in
> > > > https://github.com/submission5738/Graphs/blob/main/frequency%20convergence.png, which shows the MSE as a function of iteration. For the experiment we used two inputs, with $v_n=0.03$ and $v_n=0.5$. It can be seen that DIP converged with the latter (noisier) input faster, which induces a lower frequency target function, than with the former (less noisy) input. The reconstructed images after 150 epochs are shown in https://github.com/submission5738/Graphs/blob/main/frequency%20figure%20iteration%20150.png.
> > > >
> > > > *Experiment 2: locality bias*. Next, we design a denoising experiment to underscore the bias of real equivariant CNNs to position within the receptive field. For denoising, we feed DIP with a noisy image as input and set the loss function to recover the input (with additional noise) shifted by $s$ pixels diagonally toward the bottom-right. Note that when $s=0$ for each receptive field the central pixel should be predicted by the network, while with large $s$ excentric pixels must be predicted. Our analysis suggests that learning a map with a large shift will be slower than learning a map with a smaller shift. The experiment indeed confirms this prediction. We tested four different shifts of 0, 10, 20, and 30 pixels. The plot in https://github.com/submission5738/Graphs/blob/main/shift%20convergence.png shows MSE as a function of iteration. The figure in https://github.com/submission5738/Graphs/blob/main/shift%20figure%20iteration%20100.png shows the reconstructed image with the four shifts after 100 iterations. The difference in speed is evident in these figures.

---

### Official Review · Reviewer_sRWF · 2022-07-11

**Rating:** 7
**Confidence:** 4
**Soundness:** 3 good
**Presentation:** 3 good
**Contribution:** 4 excellent

**Summary:**

The paper studies the spectral properties of convolutional kernels that are derived from spherical kernels. The inputs are treated as multi-channel, where each channel $\zeta$ corresponds to a pixel from different patches $d$, such that the input space is the cartesian product of the $d$ hyper-spheres. The authors prove that the eigenfunctions of the studied kernels are either the product of spherical harmonics or the shift invariant sum of them, while the corresponding eigenvalue decay polynomially with the frequency of the eigenfunctions. Further, they prove that the lager eigenvalues are associated with the functions localised in the center of the receptive field of the patch. The authors conclude their analysis with evaluating the eigenvalues for the arc-cosine and the neural tangent convolutional kernels.

**Questions:**

Please see my detailed questions above under the sections of Quality and Clarity

**Limitations:**

The authors sufficiently discuss their findings and compare it to similar theoretical results in the literature.
A nice point I would like to see is the computational requirements for evaluating the kernel. That would give the reader an estimate of how feasible would be to use such kernels in practice.

**Strengths And Weaknesses:**

### Originality
This is a highly theoretical paper. Spectral analysis of the kernel functions associated with deep neural network architecture with Relu activations has received a lot of attention over the past couple of years. This paper is explicitly focusing on the analysis of the behaviour of the convolutional spherical kernels. In that spirit, the authors extended the analysis from [31] to deep kernel structures with overlapping patches. They also have sufficiently covered the related work and commented on how their work improves upon the existing literature. In my opinion the analysis is novel and provides further insights on the behaviour of the deep neural networks.

### Quality
The theoretical analysis is really good. The authors have demonstrated a high skill on deriving the end result, which I believe is significant.
My concerns are mainly with the results depicted in the plots and the quality of them. I also have some concerns regarding the presentation of the paper in certain parts.

* To be honest I am really baffled with the extremely small values of the eigenvalues in Figure 3 to the point I am questioning my understanding. Coming from the result of Theorem 3.2, the eigenvalues should decay polynomially with the frequency and the degree of that polynomial is mainly determined by $\zeta$. Given that in the computations the patch size is rather small, i.e., $\zeta=2$, why is the value of the biggest eigenvalue at the order of 1e-11? I understand that the important quantity is the ratio of the eigenvalues, but still this is a remarkably small value. It cannot be that the $\tilde{c}$ is that small, right?

* Carrying on with the small values, I would also like to understand the reason behind the really small values of the $b_n$ coefficients in Figure 2. Yes, I expect that for the value of the $\exp(50)$th coefficient will be practically 0. But what happens to the first 5-10 coefficients? Is their value sufficiently large? Is this just a scaling artifact of the plot?

Regarding the presentation:

* First of all, I believe that the authors could have been more explicit in setting up the problem. In the beginning of Section 2 there should be explicitly mentioning of what $\zeta, d$ are. My understanding is that $\mathbf{x}$ is a $\zeta \times d$ matrix, where $\zeta$ is the pixels in the patch and $d$ the number of patches. Then the input space of product of $d$ spheres makes sense.
* Despite the effort, Figure 1 is not very helpful in understanding the architecture. I had to rely on the actual maths and the shapes of the involved matrices to be sure of what is going on.
* I do not understand the inline equation of the product of spherical harmonics in section 3.1 line 123. $\mathbf{k}$ is a vector of frequencies, yet we have a product of only the $k_i$ frequency for all the patches that come after $i$. Why don't we have interactions of different frequencies between patches?
* A short discussion right after Lemma 3.1 on the connection between the result and the one from [31] would be extremely valuable for solidifying the understanding.

### Clarity
As I have already said this is a highly theoretical paper. The nature of the topic is already complicated and the required notation, together with the study of convergence bounds, do not make any favours to the reader. Having said that, the authors have done a really good effort to write and explain the paper.

Few minor remarks:
* Last paragraph in page 1 looks like having a smaller fontsize than the rest of the manuscript, please fix that.
* Page 2, line 43, on the comment around intermediate pooling. I suggest you provide further discussion on that, as the introduction of intermediate polling, apart from requiring careful handling to keep the inputs to the next layer always normalised, it also changes considerably the geometry, since it messes with the receptive field at each layer.
* Page 2, line 47. It would be wise to elaborate on what is the shift invariant sum of the spherical harmonics, as these contents have not been introduced yet.

### Significance
The theoretical results presented in the manuscript are significant and a can potentially be a good addition to the literature.

---

> ### Author Response · Authors · 2022-08-02
> **Response to Reviewer sRWF**
>
> Thank you for the constructive and encouraging remarks.
>
> *Extremely small values of the eigenvalues*: We apologize for this mistake in plotting the ticks in the Y-axis of Figure 3. The corrected figures are available at https://github.com/submission5738/Graphs/blob/main/Fig3_left.png and https://github.com/submission5738/Graphs/blob/main/Fig3_right.png. As can be seen, the first few coefficients obtain reasonable values. For example, for the CGPK, $\lambda_{1,0,0,0}=0.21$, $\lambda_{1,1,0,0}=0.011$, $\lambda_{2,0,0,0}=0.040$, $\lambda_{3,0,0,0}=0.0039$, $\lambda_{1,2,0,0}=0.0023$. We verified that this mistake only affected the figure.
>
> *Small values in Figure 2*: The values of the $b_n$’s in Figure 2 are not small – note that in this 3D plot the top values are near zero (the oblique view might be somewhat misleading). For example, the largest coefficients are $b_{0,0}=0.6$, $b_{1,0}=0.33$, $b_{2,0}=0.16$, $b_{1,1}=0.05$, $b_{5,5}=0.0016$.
>
> *Problem setting*: We regret if our setting (in lines 56-59) is unclear. Our input includes $d$ pixels and $\zeta$ channels. Starting from the second layer, stride-1 convolutions are applied to patches of size $q$ pixels.
>
> *Figure 1 is not very helpful*: Thank you for this comment. We will try to redo Figure 1.
>
> *Inline equation in line 123*: Thank you for noticing this unfortunate typo. The product should be over $i$ going from 1 to $d$.
>
> *Connection between Lemma 3.1 and [31]*: We will include the following discussion after Lemma 3.1: Lemma 3.1 generalizes a result by [31] (Theorem 4.1), which deals with the special case of a kernel made of a single convolutional layer followed by $L$ fully connected layers. Our result applies to any multi-dot product kernels, including convolutional kernels of arbitrary depths such as CNTK and CGPK.
>
> *Computational requirements for evaluating the kernel*: To compute  $k^{Eq}({\bf x},{\bf z})$ for a pair of inputs, we need first to apply $d\times\zeta$ operations to calculate the inner products between corresponding pixels. Then, in each of $L$ layers, we should apply $\kappa_1$ to averages of $q$ entries. These can be efficiently implemented in $(2d+q)L$ operations. Thus we obtain a total of $d\zeta+(2d+q)L$ operations. To compute $k^{tr}$ we need to compute $k^{eq}(s_i{\bf x},s_i{\bf z})$ (over corresponding shifts of ${\bf x}$ and ${\bf z}$) and sum these, for a total of $d(d\zeta+(2d+q)L+1)$ operations. For $k^{gap}$, we need to consider all independent shifts of ${\bf x}$ and ${\bf z}$ obtaining $d^2(d\zeta+(2d+q)L+1)$ operations.
>
> Thank you for the additional comments, which we will address.

---

> > ### Comment · Reviewer_sRWF · 2022-08-05
> > **extra clarification on $\zeta$, $d$**
> >
> > Thanks for answering most of my questions.
> >
> > I need a final clarification on the exact meaning of $\zeta$, $d$ and the product of SH.
> >
> > You are saying that $\zeta$ is the number of channels (let's assume 3 for RGB) and $d$ is the pixels. Normally we would expect a high dimensional image where $d$ is $W \times H$ of the image. Then we have a multi-sphere setup as the product of $d$ spheres $\mathbb{S}^{\zeta-1}$. Two questions/remarks:
> > 1. Each pixel lives on its own sphere and you get interactions between pixels when the corresponding eigenvalues are non-zero. Do you reduce the space of interactions by any kind of truncation?
> > 2. This set-up is practically impossible to work with. Especially the evaluation of $k^{\text{gap}}$ is a $d^3$ operation. I understand that the paper focuses on the theoretical analysis of the kernel, but how can we practically use it in real problems?

---

> > > ### Author Response · Authors · 2022-08-07
> > > **Response to additional questions**
> > >
> > > Thank you for these additional comments.
> > >
> > > We wish to emphasize that our goal in this work is to analyze the inductive bias of CNNs through their respective kernels, CNTK and CGPK, which capture the behaviour of over-parameterized CNNs. While indeed using these kernels directly is expensive, over-parameterized networks provide an efficient and practical way to utilize them in classification and regression problems.
> > > 1. For our analysis, we model all the interactions between pixels. We show these interactions are quantified by the leading coefficient of the eigenvalues. The extent of these interactions depends on the receptive field size.
> > > 2. Past work directly applied CNTK and CGPK to CIFAR-10 [2], but we are unaware of their application to larger image datasets, such as ImageNet, most probably due to their prohibitive complexity. Developing analogous kernels that can be utilized efficiently is an interesting topic for future work. This may be possible once we understand the properties of CNTK/CGPK.

---

> > > > ### Comment · Reviewer_sRWF · 2022-08-08
> > > > **Response to authors**
> > > >
> > > > Thanks for the extra clarification. I agree with your comments and I understand that it is a theoretical study of these methods. It is definitely valuable to add a paragraph discussing exactly what we have been discussing in this thread.
> > > >
> > > > In any case I remain positive about the paper and I'm inclined to increase my score to a 7.

---

> > > > > ### Author Response · Authors · 2022-08-08
> > > > > **Response to reviewer**
> > > > >
> > > > > Thank you for this response and for your positive outlook on our paper. We will certainly add text to reflect this discussion.

---

### Official Review · Reviewer_uuHF · 2022-07-13

**Rating:** 7
**Confidence:** 4
**Soundness:** 4 excellent
**Presentation:** 3 good
**Contribution:** 4 excellent

**Summary:**

The paper studies the inductive bias of deep convolutional architectures by studying spectral properties of the corresponding NNGP/GPK and NTK kernels. More specifically, the authors provide decay rates of power series coefficients and eigenvalues of various CGPK/CNTK kernels ("equivariant" where only one location is considered at the output, "trace" which corresponds to a fully-connected last layer, and "GAP", for global average pooling at the last layer). These illustrate how these architectures are biased towards certain functions that depend on few pixels, with preference for neighboring pixels.

**Questions:**

* Regarding the non-matching upper/lower bounds on eigenvalues (e.g. Theorem 3.5), could you elaborate more on which you think is tighter in which setting? It is mentioned that lower bounds may be tighter when only few pixels are considered, while upper bounds are tighter when more pixels are considered -- can this be formalized more precisely?

* In corollary 3.8, for the Tr and GAP kernels, isn't $p_i$ the same for all $i$, given the symmetry of the kernels? If so, is there a better way to illustrate positional biases w.r.t. relative positions?

* Can the arguments on number of GD iterations (paragraph starting at line 226) be made more formal? It would be good to have cleaner propositions for this, as it is not immediate from the statements on eigenvalues.

* [This paper](https://arxiv.org/abs/2112.05611) seems closely related to the present manuscript and the authors should compare to it.

**Strengths And Weaknesses:**

The paper is well written, and the problem studied is significant, as it provides new insight on the role of convolutional architectures in deep models. The theory is sound and the results are clearly positioned compared to other works in the area.

Certain results are not fully tight, for instance some lower and upper bounds on eigenvalues are non-matching, which suggests that a tighter theory might be developed. However, this might be difficult to achieve given the complicated form of the studied kernels, so I believe the current results are already good enough for acceptance.

---

> ### Author Response · Authors · 2022-08-02
> **Response to Reviewer uuHF**
>
> Thank you for the thoughtful and encouraging remarks.
>
> *Non-matching upper/lower bounds*: Achieving a tight bound indeed appears to be difficult due to the complicated form of the kernels. Our numerical experiments (Figure 2) indicate that the Taylor coefficients lie near the lower bound for terms that involve a single pixel, whereas they lie near the upper bound for terms that involve all pixels. The same behavior is observed also for the eigenvalues, as is apparent from Figure 3(left). For this plot ($d=4,\zeta=3$), when the frequency varies in one pixel the eigenvalues should decay according to Theorem 3.5 between $k^{-5}$ and $k^{-2.75}$. The plot shows $k^{-5.6}$, close to the lower bound of $k^{-5}$ (the slope slowly increases and converges to $k^{-5}$). In contrast, when the frequency varies in all 4 pixels, the eigenvalues should decay between $k^{-20}$ and $k^{-11}$. Here we obtain a decay of $k^{-11.5}$, close to the upper bound.
>
> *Positional bias for the Tr and GAP kernels*: We apologize for the lack of clarity. $p_i$ in Corollary 3.8 denotes the number of paths from input node $i$ to the *equivariant kernel*. Therefore, $p_i$ varies with the eccentricity of $i$ in the receptive field of this kernel. The eigenvalues of the trace and GAP kernels are determined by averaging the corresponding eigenvalues of the equivariant kernel over all shifts of indices (Theorem 3.7). This averaging makes the eigenvalues shift invariant; therefore, the eigenvalues for the trace and GAP kernels depend on relative positions. We will rephrase Corollary 3.8 to clarify the definition of $p_i$.
>
> *Number of GD iterations*: For this argument we rely on Basri et al. 2019 [6]. In short, Arora et al. (ICML 2019) showed that for a training set $X \in R^{d \times n}$ and target values $Y \in R^n$ we have that $||f^{(t)}(X)-Y|| \approx ||(I-\eta K)^tY|| = \sqrt{\sum_{i=1}^n(1-\eta \lambda_i)^{2t}(v_i^T Y)^2}$, where $f^{(t)}(X)$ is the network prediction for all training samples after $t$ iterations, $K$ is the NTK kernel matrix, $\lambda_i,v_i$ are its eigenvalues and eigenvectors (these converge to the eigenvalues and eigenfunctions of NTK in the limit of a training set with infinitely many samples that are distributed uniformly), $\eta$ is the learning rate,and $I$ denotes the identity matrix. Consequently, suppose $Y$ is an eigenfunction, then to achieve accuracy $\epsilon$, i.e., $(1-\eta \lambda_i)^t<\epsilon$, we obtain that $t>-\log(\epsilon)/(\eta \lambda_i)=O(1/\lambda_i)$.
>
> *Related reference by Xiao*: Thank you for this related reference. Our paper differs from this paper in several significant respects. Most importantly, while our analysis addresses the case of stride-1 convolutions, i.e., when the network processes information from overlapping patches, Xiao’s paper addresses only the case of convolutions with non-overlapping patches (i.e., stride-$q$ when $q$ is the patch size). Secondly, unlike our paper which handles the case of constant input dimension and varying frequency, Xiao’s paper analyzes the case of constant frequency and varying (growing) input dimension. Finally, their setting does not allow for ReLU activation.

---

### Meta-Review · Area_Chair_3fpV · 2022-08-23

**Recommendation:** Accept
**Confidence:** Certain

**Metareview:**

The paper received positive and borderline reviews. A few technical concerns were raised but the rebuttal has addressed those convincingly. There is a clear consensus that the paper makes a valuable theoretical contribution and that it should be accepted. The AC agrees with the reviewer's assessment and follows their recommendation.

**Award:**

No

---

### Decision · Program_Chairs · 2022-09-14

Accept